

# An automatic observation-based typing method for EARLINET

Nikolaos Papagiannopoulos[1,2], Lucia Mona[1], Aldo Amodeo[1], Giuseppe D'Amico[1], Pilar Gumà Claramunt[1], Gelsomina Pappalardo[1], Lucas Alados-Arboledas[3,4], Juan Luís Guerrero-Rascado[3,4], Vassilis Amiridis[5], Arnoud Apituley[6], Holger Baars[7], Anja Schwarz[7], Ulla Wandinger[7], Ioannis Binietoglou[8], Doina Nicolae[8], Daniele Bortoli[9], Adolfo Comerón[2], Alejandro Rodríguez-Gómez[2], Michaël Sicard[2,10], Panagiotis Kokkalis[11,12], Alex Papayannis[12], and Matthias Wiegner[13]

[1]Consiglio Nazionale delle Ricerche, Istituto di Metodologie per l'Analisi Ambientale (CNR-IMAA), C.da S. Loja, Tito Scalo (PZ), 85050, Italy
[2]CommSensLab, Dept. of Signal Theory and Communications, Universitat Politècnica de Catalunya, Barcelona, Spain
[3]Andalusian Institute for Earth System Research (IISTA-CEAMA), 18006, Granada, Spain
[4]Department of Applied Physics, University of Granada, 18071, Granada, Spain
[5]IAASARS, National Observatory of Athens, Athens, Greece
[6]Royal Netherlands Meteorological Institute KNMI, De Bilt, the Netherlands
[7]Leibniz Institute for Tropospheric Research (TROPOS), Leipzig, Germany
[8]National Institute of R&D for Optoelectronics (INOE), Magurele, Romania
[9]Earth Science Institute-(ICT), Évora, Portugal
[10]Ciències i Tecnologies de l'Espai - Centre de Recerca de l'Aeronàutica i de l'Espai/Institut d'Estudis Espacials de Catalunya (CTE-CRAE/IEEC), Universitat Politècnica de Catalunya, Barcelona, Spain
[11]Laser Remote Sensing Unit, Physics Dept., National Technical University of Athens, Athens, Greece
[12]Physics Department, Faculty of Science, Kuwait University, Kuwait
[13]Ludwig-Maximilians-Universität (LMU), Meteorologisches Institut, Theresienstraße 37, 80333 Munich, Germany

*Correspondence to:* Nikos Papagiannopoulos (nikolaos.papagiannopoulos@imaa.cnr.it)

**Abstract.** We present an automatic aerosol classification method based solely on European Aerosol Research Lidar Network (EARLINET) intensive optical parameters with the aim of building a network-wide classification tool that could provide near-real-time aerosol typing information. The presented method depends on a supervised learning technique and makes use of the Mahalanobis distance function that relates each un-classified measurement to a pre-defined aerosol type. As a first step (training

5   phase), a reference dataset is set up consisting of already classified EARLINET data. Using this dataset, we defined eight aerosol classes: clean continental, polluted continental, dust, mixed dust, polluted dust, mixed marine, smoke, and volcanic ash. The effect of the number of aerosol classes has been explored, as well as the optimal set of intensive parameters to separate different aerosol types. Furthermore, the algorithm is trained with literature particle linear depolarization ratio values. As a second step (testing phase), we apply the method to an already classified EARLINET dataset and analyse the results of the

10  comparison to this classified dataset. The predictive accuracy of the automatic classification varies between 59 % (minimum) and 90 % (maximum) from 8 to 4 aerosol classes, respectively, when evaluated against pre-classified EARLINET lidar. This indicates the potential use of the automatic classification to all network lidar data. Furthermore, the training of the algorithm with particle linear depolarization values found in literature further improves the accuracy: the accuracy range is 69–93 % from 8 (minimum) to 4 (maximum) aerosol classes, respectively. Additionally, the algorithm has proven to be highly versatile as it



adapts to changes in the size of the training dataset and the number of aerosol classes and classifying parameters. Finally, the low computational time and demand for resources make the algorithm extremely suitable for the implementation within the Single Calculus Chain (SCC), the EARLINET centralised processing suite.

## 1 Introduction

The European Aerosol Research Lidar Network (EARLINET; Pappalardo et al., 2014) operates Raman lidars at a continental scale. Since the beginning, the network aimed towards a sustainable observing system that has been achieved by developing a quality assurance strategy, and optimizing instruments and data. To this direction and towards future advancement, the network plans continuous measurements and near-real-time data delivery. With this in mind, the Single Calculus Chain (SCC; D'Amico et al., 2015) for the automatic lidar analysis has been developed and currently delivers profiles of optical aerosol properties. The

EARLINET SCC explores the implementation of new features like profiles of intensive optical properties and determination of aerosol layer geometrical properties. The intensive optical properties are type-dependent and can be used to classify the observed layers into aerosol types. The categorization into different types provides significant help to understand aerosol sources, their effects, and feedback mechanisms to improve the accuracy of satellite retrievals and to quantify assessments of aerosol radiative impacts on climate (Russell et al., 2014) by intercomparing numerical models such as NWP (Numerical Weather

Prediction) and CTM (Chemical Transport Model) (Baklanov et al., 2014). Thus, EARLINET by providing multi-wavelength range resolved aerosol properties, has an added value for aerosol typing. In this study we present a flexible automatic method to classify EARLINET data.

Lidar systems are capable of identifying multiple layers in the atmosphere owing to their high vertical resolution (on the orders of tens of meters). Thus, lidar-based retrievals can provide a separate classification for each layer and are not confined to

columnar classifications as in the case of sunphotometers. The lidar technique has proven to be a robust tool to classify aerosols with its capability of polarization-sensitive and multi-wavelength measurements (Liu et al., 2008). Sophisticated lidars, such as the High Spectral Resolution Lidars (HSRL) and the multi-wavelength Raman lidars, offer a multitude of intensive parameters that characterize different aerosol types (e.g., Müller et al., 2007a; Burton et al., 2012; Groß et al., 2013). Typically, the particle extinction-to-backscatter ratio (i.e., particle lidar ratio), the particle linear depolarization ratio at one or more wavelengths

and the wavelength dependence of extinction and/or backscatter coefficients (i.e., extinction- or backscatter-related Ångström exponents) are considered.

The increasing amount of available information and particularly the plethora of lidar intensive parameters, can offer a more accurate aerosol classification as well as insight into the various aerosol types (Burton et al., 2013). Consequently, objective, multivariate analysis is needed to take advantage of this information. Automatic algorithms are, therefore, employed to classify

aerosol into respective types. These procedures make use of various classifiers that are able to quantify the differences between the aerosol classes. In classification analysis, the observations are allocated to a known number of groups, that is, a supervised learning technique. Whereas in cluster analysis, neither the number of groups nor the groups are known beforehand and the classifier is tasked with it.





The measured values are evaluated by the classification function to find the group to which the individual most likely belongs. Specifically, distance-based classification techniques (e.g., $k$- nearest neighbour, support vector machine algorithms) are straightforward, that is, the classification depends on the distance from the target instance to the training instance. The Mahalanobis distance classifier (Mahalanobis, 1936) has a wide range of applications and can be used to categorize data points,

each representing an observation, into classes that have predefined characteristics. The distances between the observation and the different classes are calculated, and then the observation is attributed to the class for which the distance is the minimum.

The Mahalanobis-distance-based classification found great applicability in aerosol studies. For instance, the algorithm developed by Burton et al. (2012) makes use of four lidar intensive properties, namely the particle linear depolarization ratio at 532 nm, the particle lidar ratio at 532 nm, the backscatter-related 532-to-1064- nm color ratio, and the ratio of particle linear

depolarization ratios at 1064 and 532 nm in order to classify aerosols into eight types. A slightly different algorithm including also the uncertainties on the input properties was introduced by Russell et al. (2014). Their algorithm was applied to satellite derived optical and physical data. The reference dataset was obtained from AERONET (Aerosol Robotic Network; Holben et al., 1998) stations where a single aerosol type tends to dominate (e.g., Cattrall et al., 2005). The pre-specified classes were then applied to a five-year record of retrievals from the spaceborne POLDER 3 (Polarization and Directionality of the Earth's

Reflectances 3; Tanré et al., 2011) polarimeter on PARASOL (Polarization and Anisotropy of Reflectances for Atmospheric Sciences coupled with Observations from a Lidar; Tanré et al., 2011) spacecraft. Recently, Hamill et al. (2016) used the same classifier to produce an aerosol classification scheme based on long term AERONET data.

In this work, we present a method analogous to the one proposed by Burton et al. (2012), modified to fit EARLINET's needs and capabilities. The aerosol typing exclusively makes use of EARLINET lidar-derived intensive property data. We

use the Mahalanobis distance as a classifier to assign any given multi-dimensional observation to the pre-specified aerosol class to which it is most similar. These classes are defined using an EARLINET-based classification scheme. The EARLINET classification scheme is presented in Sect. 2 where we also describe the parameters readily delivered by the network that can be used to classify aerosols. Furthermore, the major aerosol types that comprise the aerosol classes onto which the aerosol classification is based are presented. In Sect. 3 the method that we apply to EARLINET data is explained, and, we present

the training phase. We set up a scheme for investigating the number of aerosol classes and we perform an analysis to identify the intensive parameters that contribute the most to the classification as well. Sect. 4 describes the testing phase and provides a discussion of the results of the classification. The paper closes with conclusions of our study and suggestions for further applications and improvements.

## 2  Operational network–EARLINET

EARLINET (www.earlinet.org) was established in 2000, providing aerosol profiling data on a continental scale, and now is part of the Aerosols, Clouds, and Trace gases Research InfraStructure (ACTRIS; www.actris.eu/). In these 18 years of continuous existence, EARLINET has evolved both in the number of contributing stations, as well as in its observing capacity (Pappalardo et al., 2014). Currently, 30 stations are submitting aerosol extinction and/or backscatter coefficient profiles to the



EARLINET database, according to EARLINET's measurement schedule (one daytime and two nighttime measurements per week). Therefore these systematic observations consolidate a 4D European quantitative and statistically significant aerosol survey. Further measurements are devoted to special events, such as volcanic eruptions, forest fires, and desert dust outbreaks. Moreover, EARLINET provides correlative measurements during CALIPSO (Cloud-Aerosol Lidar and Infrared Pathfinder

Satellite Observations) overpasses on each EARLINET station in order to validate satellite products (e.g., Mamouri et al., 2009; Mona et al., 2009).

The majority of the EARLINET stations (67 % of the stations; Pappalardo et al., 2014) operates multi-wavelength Raman lidars that combine a set of elastic and nitrogen inelastic channels, typically consisting of three elastic and two inelastic Raman channels (the so-called $3\beta + 2\alpha$ configuration). In particular, they provide the aerosol extinction (at 355 nm and 532 nm),

and backscatter coefficients (at 355 nm, 532 nm, and 1064 nm). This configuration allows the retrieval of the range-resolved particle lidar ratio at 355 nm and 532 nm $- S_{aer}^{\lambda}$. This intensive parameter depends on the shape, size, and chemical composition of the aerosol (Müller et al., 2007a). When lidar ratio is available for more than one wavelength, the corresponding color ratio can be also retrieved $- S_{aer}^{\lambda_1}/S_{aer}^{\lambda_2}$. This quantity is a robust means to characterize the ageing status of smoke particles as well as the spectral dependence of aerosol (Müller et al., 2007a; Alados-Arboledas et al., 2011; Nicolae et al., 2013; Pereira et al.,

2014; Samaras et al., 2015). The combination of the optical data allows the retrieval of the size sensitive backscatter and/or extinction related Ångström exponent and can be calculated as

$$\kappa_X(\lambda_1, \lambda_2) = \frac{ln[X(\lambda_1)/X(\lambda_2)]}{ln(\lambda_2/\lambda_1)} \tag{1}$$

with $X$ denoting the backscatter ($\beta$) or extinction coefficient ($\alpha$) for a set of wavelengths, $\lambda_1$ and $\lambda_2$. Moreover, 52 % of EARLINET stations (Pappalardo et al., 2014) are equipped with depolarization channels, thus providing profiles of the particle

linear depolarization ratio. It can be calculated according to Biele et al. (2000); Freudenthaler et al. (2009)

$$\delta_{aer}^{\lambda} = \frac{(1+\delta_m)\delta_v R - (1+\delta_v)\delta_m}{(1+\delta_m)R - (1+\delta_v)} \tag{2}$$

with $R$ the backscatter ratio, $\delta_m$ the molecular depolarization, and $\delta_v$ the volume depolarization ratio. This parameter provides information on the particle shape, thus enhancing the aerosol typing strength of the network. Under favourable conditions, the aerosol microphysical properties, such as the effective radius, the volume concentration and the refractive index can also be

retrieved through complex numerical algorithms (e.g., Müller et al., 2004; Veselovskii et al., 2010; Bovchaliuk et al., 2016; Chaikovsky et al., 2016).

The data products described above make the EARLINET data an excellent basis to perform aerosol typing at continental scale. Examples of methodologies to classify aerosol datasets can be found in e.g., Müller et al. (2007a, b); Groß et al. (2011); Mona et al. (2012a); Navas-Guzmán et al. (2013); Baars et al. (2016). At the time being there are different algorithms under

development which combine measurements and aerosol models (Nicolae et al., 2016; Wandinger et al., 2016). Nevertheless, automated observation-based algorithms working at network level for the identification of layers, their boundaries, and the



corresponding aerosol typing are not yet available. The SCC tool for automatic processing of EARLINET lidar signals is, currently, providing primarily profiles of particle extinction and backscatter coefficients and volume and particle depolarization ratio. The SCC aims at incorporating modules for layer identification, intensive properties retrieval, and aerosol typing. Therefore, this paper could provide a starting point for a harmonized EARLINET classification tool that could also be used

by other lidar networks, like the ones involved in GALION (GAW Aerosol Lidar Observation Network), the GAW (Global Aerosol Watch) initiative for the aerosol lidar observation on a global scale, and within aerosol lidar studies in general.

## 2.1  EARLINET manual aerosol classification

The typical procedure for aerosol categorization adopted within the EARLINET community consists of three main steps:

1. layer identification and cloud screening,

2. identification of the geometrical properties (boundaries, center of mass) of the aerosol layer, and

3. the aerosol layer typing by means of investigation of intensive optical properties (Ångström exponents, lidar ratios, and particle linear depolarization ratios), model outputs (backward trajectory analyses), and ancillary instruments data if available (e.g., satellite or sunphotometer data).

In what follows, an example of an aerosol type assignment using EARLINET data is presented. Figure 1 shows the temporal

evolution of the range corrected signal at 1064 nm from a measurement made in Potenza, Italy, on 14 July 2011, 19:20–22:10 UTC with the reference lidar system MUSA (Multiwavelength system for aerosol) of CNR-IMAA (Consiglio Nazionale delle Ricerche - Istituto di Metodologie per l'Analisi Ambientale). High values show a stratified aerosol layer from the ground up to 5 km, whereas low values indicate aerosol free regions. The lowest altitude range presents the overlap between the laser beam and the receiver field of view and, therefore, it is the blind range of the lidar. The optical thicker layer lies below 2 km,

with a distinct layer atop extending up to 3.5 km and, finally, an optically thinner region from 3.5 km to 5 km. The retrieved profiles for the same temporal window of particle backscatter and extinction coefficient, lidar ratios, and Ångström exponents are shown in Fig. 2. The aforementioned layers present the same behavior as seen in the intensive optical profiles and, thus, correspond to the same type of particles. The mean values of all optical parameters in the range 1.6-5 km are calculated: lidar ratios of $53 \pm 7$ sr at 355 nm and $55 \pm 8$ sr at 532 nm and Ångström exponents of 0–0.4 were found. These values indicate the

presence of coarse particles and they are in accordance with the typical dust values observed over Potenza (Mona et al., 2014).

For the classification of aerosols with respect to their source regions and age, auxiliary information like results of transport and dispersion models or satellite data are used. For the observed aerosol layer, the Lagrangian dispersion model FLEXPART (FLEXible PARTicle dispersion model; Stohl et al., 2005) model was used for a 5-day backward simulation. Figure 3 shows the so-called footprint that indicates the areas of the air parcels travelling below 2 km before reaching the study area. The model

output is given in terms of the decimal logarithm of the integrated residence time in seconds in a grid box. The most probable aerosol source region and the aerosol type were assigned accordingly. The dust-prone area of northern Africa (Morocco and



northern Algeria) is most likely the source of the particles and in conjunction with the lidar-derived information the inferred type is pure dust.

In the following, the characteristics of the major aerosol types are presented. These aerosol types are used for the automatic classification and correspond to aerosol layers typically encountered over Europe.

## 2.2  Aerosol Types

One of the defining characteristics of the aerosol properties is the source; aerosols found in the atmosphere can be, for example, mineral particles from arid areas of the Earth or organic carbon emitted during biomass burning events. Due to the multiple influence of the aerosol origin on the properties, aerosol sources can be used to classify them into different categories. In this section, we provide an overview of the main aerosol types observed over the EARLINET stations followed by the corre-
sponding optical properties. This section also aims to provide the important information of the aerosol types that the automatic classification is based upon. The considered aerosol types almost coincide with the ones used in the CALIPSO classification scheme (Omar et al., 2009) which provides already a satisfactory description of the atmospheric aerosol content. Moreover, adopting similar classification schemes, the direct comparison of the proposed typing against the CALIPSO product is possible.

### 2.2.1  Continental

Man-made activities dictate the aerosol pattern within the atmospheric boundary layer, and affect the observations in the lower troposphere in Europe. Anthropogenic particles show a strong wavelength dependence of their optical properties, i.e., high Ångström exponent values. Moreover, they are typically small and do not significantly depolarize the backscattered light ($\delta_{aer}^{532} = 0.04 \pm 0.04$; Heese et al., 2016), and due to the high carbon content, these particles reveal high lidar ratios (Giannakaki et al., 2010). We refer to this particle type, herein, as polluted continental.

Typically, the clean continental aerosol over Europe is a mixture of anthropogenic pollution with particles from natural sources. The clean continental type shows low depolarizing ability with values lower than 0.07 (Omar et al., 2009), low lidar ratio values, i.e., 20–40 sr and relatively high Ångström exponents, i.e., 1.0–2.5 (Ansmann et al., 2001; Giannakaki et al., 2010). The clean continental, therefore, differentiates from the polluted continental type due to the less light absorbing properties.

### 2.2.2  Marine

Marine particles are produced at the sea surface and dominate the shallow boundary layer over the oceans (e.g., O'Dowd and de Leeuw, 2007). Specifically, the sea-salt particles feature a predominant coarse mode, however, they are spherical in humid conditions and weakly absorbing in contrast to the dust particles. Therefore, they yield low particle lidar ratio values, are almost non depolarizing and exhibit low Ångström exponent values (e.g., Burton et al., 2014; Dawson et al., 2015). This aerosol type is mainly identifiable by the low particle lidar ratio, i.e., 15–25 sr at 532 nm (Burton et al., 2012). As marine aerosol
layers manifest themselves over water bodies, either only stations located at the shorelines and under specific meteorological conditions or shipborne measurements can observe pure maritime particles. Consequently, the observation of pure maritime





particles is rare within EARLINET and, generally, when these particles are observed their characteristics are far from pristine (Preißler et al., 2013; Papagiannopoulos et al., 2016a). However, mixtures with important contribution of marine particles can be observed in the Mediterranean basin (Papagiannopoulos et al., 2016a).Thus, we consider pure marine and marine dominated layers as one single category denoted as mixed marine.

### 2.2.3 Mineral dust and dust mixtures

Mineral dust is produced in arid and semi arid regions of the world, and has a profound contribution to the total natural aerosol loading (Ginoux et al., 2001). The optical properties are considerably different from the other types, thus making them easy to identify. The irregular shape and the large size ($< 50 \mu m$; Mahowald et al., 2014) lead to a significant high depolarization of the backscattered radiation (e.g., $\delta_{aer}^{532} = 0.34 \pm 0.02$ for Saharan dust over Germany, Wiegner et al., 2011), and to medium lidar ratio values (e.g., $S_{aer}^{532} = 55 \pm 10$ sr, Tesche et al., 2013; Mona et al., 2014). They are spectrally neutral to backscatter and extinction, and thus produce low Ångström exponent values (Wiegner et al., 2011). Therefore, the particle lidar ratio, particle linear depolarization ratio, and the Ångström exponent are excellent physical parameters to characterize mineral dust and to distinguish it from other aerosol types. However, it needs to be taken into account that the dust optical properties depend on the source region and the transport pattern (Valenzuela et al., 2014), which is a source of variability mainly detected in the lidar ratio (e.g., Schuster et al., 2012; Nisantzi et al., 2015). Recently, Mamouri et al. (2013) showed that dust originating from the Arabian desert produced significantly lower lidar ratio values (34–39 sr at 532 nm) than respective values (50–60 sr at 532 nm) from western Saharan dust particles. An overview on the dust characterization using lidar measurements can be found in Mona et al. (2012b).

Dust can be transported over continental scales. In particular, Saharan dust outbreaks to Europe and across the Atlantic Ocean have been deeply investigated. The European continent is regularly influenced by advected Saharan particles as has been discussed by e.g., Ansmann et al. (2003); Guerrero-Rascado et al. (2008, 2009); Papayannis et al. (2008); Müller et al. (2009); Córdoba-Jabonero et al. (2011); Preißler et al. (2011); Valenzuela et al. (2012); Papayannis et al. (2014); Binietoglou et al. (2015); Bravo-Aranda et al. (2015); Granados-Muñoz et al. (2016a). The study of Papayannis et al. (2008) indicated a large variability of the measured lidar ratio and Ångström exponent values among the different sites, suggesting mixing at different levels. Additionally, the mixture processes also produce large variability of intensive properties as measured at the same site (e.g., Mona et al., 2006, 2014). As a consequence of the complex structure of the observed aerosols over Europe and the effects of transport and mixing on the properties of these particles we consider the use of three dust groups: pure dust, mixed dust and polluted dust. The pure dust group refers to particles for which the mixing with other aerosol types is negligible. Mixed dust refers to dust dominated layers mixed with marine particles. This leads to less depolarizing, and less absorbing particles with respect to pure dust particles. Papagiannopoulos et al. (2016a) found this mixture to be important in the Mediterranean region and suggested its inclusion in the CALIPSO retrieval scheme for improving the accuracy of aerosol backscatter and extinction coefficient profiles. Finally, the polluted dust category consists of dust dominated mixtures with smoke and/or continental, which produce lower depolarization, higher lidar ratios and enhanced Ångström exponent values





owing to the presence of small, spherical particles (Groß et al., 2011; Burton et al., 2012; Tesche et al., 2013; Bravo-Aranda et al., 2015).

### 2.2.4 Smoke

Biomass burning is a major global source of atmospheric aerosols. Generally, smoke particles are relatively small, spherical, and highly absorbing that produce low depolarization, high Ångström exponents, and large lidar ratios (Amiridis et al., 2009; Baars et al., 2012; Giannakaki et al., 2016). The optical properties of smoke particles may vary due to the vegetation type of the emitting source, the combustion type (smouldering or flaming fires), and atmospheric conditions (e.g., Balis et al., 2003). Furthermore, the particles are susceptible to changes during their lifetime in the atmosphere (Nicolae et al., 2013). Several EARLINET-based studies have focused on observations and characterization of smoke plumes (e.g., Müller et al., 2005; Papayannis et al., 2008; Ansmann et al., 2009; Tesche et al., 2011; Alados-Arboledas et al., 2011; Pereira et al., 2014; Ancellet et al., 2016; Ortiz-Amezcua et al., 2017), demonstrating that it is a frequently encountered aerosol type over Europe. In particular, biomass burning aerosol originating from forest fires in Canada and Siberia is regularly observed between May and October (Amiridis et al., 2009; Sicard et al., 2012a; Ortiz-Amezcua et al., 2017). However, the similarities of the physical characteristics of smoke particles and continental particles result in similar optical properties, making these types difficult to distinguish. In this work, biomass burning particles are treated as a single category called smoke.

### 2.2.5 Volcanic ash

Volcanoes are another important source of atmospheric aerosols. Volcanic eruptions eject great amounts of material in the atmosphere (tephra), while the fraction smaller than 2 mm is labeled as volcanic ash. Most of these aerosols will settle only a few tens of kilometres away from the volcano but smaller particles can travel thousands of kilometres and affect wider areas (Mattis et al., 2010; Sawamura et al., 2012; Sicard et al., 2012b; Navas-Guzmán et al., 2013; Kokkalis et al., 2013; Pappalardo et al., 2013). The optical properties of volcanic ash aerosols is generally similar to the one of desert dust, as was shown by Ansmann et al. (2010) and Wiegner et al. (2012) for fresh ash with particle linear depolarization ratios reaching 0.37 and lidar ratios of 50–65 sr. Aged volcanic particles as observed by Papayannis et al. (2012) indicate less non-sphericity with depolarization ratio values of 0.1–0.25 and lidar ratios for 355 nm within the range 55–67 sr and for 532 nm 76–89 sr. More details can be found in Mona and Marenco (2016) where the authors give a summary of how the intensive optical properties vary as a function of time. Furthermore, volcanic eruptions inject sulfur dioxide into the atmosphere thus leading to sulfate particles. Pappalardo et al. (2004) and Wang et al. (2008) reported lidar ratios of 50–60 sr and backscatter-related Ångström exponent of 2.7, signature of sulfate particles originating from Mount Etna, Italy. Moreover, CALIPSO measurements indicated low particle depolarization ratio for sulfate-rich volcanic clouds (Prata et al., 2017). Consequently, the difference in the optical properties make lidar a powerful tool for volcano monitoring. However, in this study sulfate particles and aged volcanic particles are not considered. The aerosol type relevant to the airborne ash refers to fresh ash and is denoted as volcanic ash.





# 3 Automatic aerosol type classification

## 3.1 Methodology

We developed an automated typing method, based on the work of Burton et al. (2012), but modified it in order to be compatible with the database of EARLINET. We used EARLINET data from a $3\beta + 2\alpha$ setup, hence, data retrieved during nighttime

operation. Doing this, we can estimate the maximum number of intensive parameters relevant for the classification and, furthermore, potentially apply the classification to historical EARLINET data. Besides, we explore the inclusion of depolarization ratio observations in the automatic algorithm. The selection of the aerosol types is based on the major aerosol components found over EARLINET sites and it is, also, examined the possibility of combining aerosol types to obtain better results.

The method can be separated into two important steps: the training (Sect. 3.2) and the testing (Sect. 4.1) phase. The first step

consists of the following procedures. As described in Sect. 3.2.1, well characterized aerosol samples are manually separated into classes based on their physical characteristics and this presents the reference dataset. This procedure involves the determination of each observed aerosol layer location and the estimation of mean layer intensive optical properties. Based on this analysis, the classifying parameters that provide the adequate information for a better discrimination of the aerosol type are selected (Sect. 3.2.2). Next, in order to estimate how accurately a predictive model will perform, the reference dataset is split into

training and validation datasets and the application of the classifier is evaluated (Sect. 3.2.3). Sect. 3.2.4 describes the inference of characteristic depolarization values in the algorithm with the intention to increase the prediction of the model.

For the second step, unclassified EARLINET data (the testing dataset) is categorized using the reference dataset, that is, aerosol layers are identified and their mean layer intensive properties are entered in the automatic classification scheme. Besides, these data have been classified following the method shown in Sect. 2.1 and, hence, compared against the output of the

automatic procedure. Figure 4 illustrates the sequence of the proposed methodology starting from the setting of the training dataset up to the assessment of the learning success during the testing phase.

Distance-based classification methods aim to assign an observation to a particular class based on the distance of the observation from each class center. In general, the Mahalanobis distance between an observation $x = (x_1, ..., x_p)^t$ and the mean class $\bar{x} = (\bar{x}_1, ..., \bar{x}_p)^t$ in the p-dimensional space $\mathbb{R}^p$ is defined as

$$\boldsymbol{D}_M(x, \bar{x}) = \sqrt{(\boldsymbol{x} - \bar{\boldsymbol{x}})^T \boldsymbol{S}^{-1} (\boldsymbol{x} - \bar{\boldsymbol{x}})}. \tag{3}$$

Where $\boldsymbol{S}$ is the class covariance matrix. The surfaces identified by the equation $D_M = const.$ are ellipsoids that are centered around the mean $\bar{x}$. The main characteristic of the multivariate Mahalanobis distance is that it accounts for the variance of each variable and the covariance between variables. By contrast, the Euclidean distance treats all the variables in the same way and the constant distance surfaces from a fixed point are represented by a sphere.

The Mahalanobis distance of an observation from an aerosol class is estimated, and it is assigned to the aerosol class for which the distance is minimum. The minimum accepted distance is set to a threshold assuming that the statistical distance belongs to a chi-squared distribution. The selected threshold distance represents the 99.9 % cumulative probability contour



of the class distribution and varies with the degrees of freedom (i.e., the classifying parameters). In this work, the minimum distance is 4 and corresponds to a 3-dimensional space, considering 3 classifying parameters. Moreover, the estimated Mahalanobis distances for each class can be assigned to a probability. These probabilities are normalized using the sum of each class probability and a measurement point is labelled if the relative probability is greater than 50 %. Otherwise, the type assignment

is difficult as the measurement can be equidistant from 2 or more aerosol type classes, and possibly indicate the mixing of these aerosol types. In the future, this information could be used to label complicated aerosol scenes with different level of aerosol mixing.

## 3.2 Training phase

### 3.2.1 Dataset

In supervised learning techniques, the reference dataset is crucial to the overall predictive performance of the algorithm. Therefore, it is fundamental to use well-characterized EARLINET profiles. Namely, EARLINET aerosol classified layers from Pappalardo et al. (2013); Papagiannopoulos et al. (2016a); Schwarz (2016) were used and will be presented below.

EARLINET observations from 2008 to 2010 were analyzed and the aerosol types were determined with respect to the source origin following a similar approach to Sect. 2.1 (Schwarz, 2016) and present the backbone of the reference dataset.

Table 1 lists the classified aerosol types of the above study (644 individual aerosol layers) with respect to the aerosol types presented in Sect. 2.2; however, all these aerosol layers cannot be used given the need for the maximum optical properties available (column "only from $3\beta + 2\alpha$"). The mixtures category includes all the mixtures of two or more aerosol species without containing polluted dust and mixed dust categories that are reported individually.

As discussed above, the requirement for $3\beta + 2\alpha$ lidar configuration pinpoints the low occurrence (see Table 1) of some

20 aerosol types such as the clean continental, polluted dust, and dust. Furthermore, marine aerosol was not reported in the study and the volcanic layers do not reflect the volcanic ash characteristics described in Sect. 2.2. Conversely, the latter were volcanic layers found in the stratosphere and, thus, different from the fresh ash that we consider. In order that the aerosol classes include all the major aerosol components, the aforementioned aerosol types need to be enhanced with other observations. Therefore, we implemented EARLINET network-wide typing results already published in literature (Pappalardo et al.,

2013; Papagiannopoulos et al., 2016a) for a total of 69 samples, as the reference dataset. Note that calibrated particle linear depolarization ratio profiles are not available in the selected dataset.

The type-dependent mean properties are reported in Table 2 and coincide with the typical values as of those in Sect. 2.2. Specifically, dust and volcanic types present the same characteristics with Ångström exponent as low as 0, although dust lidar ratio is $58 \pm 12$ and $55 \pm 7$ sr, for 355 nm and 532 nm respectively, and is higher than the volcanic lidar ratio ($S_{aer}^{355}=50 \pm 11$ and

30 $S_{aer}^{532}=48 \pm 13$ sr). The Ångström exponent for mixed dust is between 0.4–0.7 and lidar ratio values are below 50 sr, whereas for polluted dust the Ångström exponent lies within 0.6–1.0 and lidar ratio values for 355 nm and 532 nm are $54 \pm 8$ and $64 \pm 9$ sr. This behavior reflects the mixing of dust with pollution/smoke that tends to decrease the size of the aerosol mixture and increase its absorbing capacity. Polluted continental and smoke reveal the same size characteristics with Ångström exponent

(c) Author(s) 2018. CC BY 4.0 License.





around ∼1.3. The smoke mean lidar ratio values present the higher ones among the aerosol types – i.e., $81\pm16\,\mathrm{sr}$ and $78\pm11\,\mathrm{sr}$ – and the polluted continental values succeed with $69\pm12\,\mathrm{sr}$ and $63\pm13\,\mathrm{sr}$ for $355\,\mathrm{nm}$ and $532\,\mathrm{nm}$ respectively. For clean continental the Ångström exponent is between 1.0–1.7 and lidar ratios, for $355\,\mathrm{nm}$ and $532\,\mathrm{nm}$, are $50\pm8\,\mathrm{sr}$ and $41\pm6\,\mathrm{sr}$. This characteristic separates clean continental from polluted continental as the particles are less absorptive. Finally, mixed marine

particles are found to be relatively small in size with Ångström exponents in the range 0.8–1.0 and, thus, overlap with other aerosol types. The characteristic parameter that defines the mixed marine category is the lidar ratio, the values are found the smallest – $S_{aer}^{532}=24\pm8\,\mathrm{sr}$ – among the aerosol types.

In the proposed method, the aerosol layers are classified in terms of the aerosol types described in Sect. 2.2. As a starting point for this study, we use 8 aerosol classes: clean continental (CC), polluted continental (PC), pure dust (D), mixed dust

(MD=Dust+Marine), polluted dust (PD=Dust+Smoke and/ or Dust+Polluted Continental), mixed marine (MM), smoke (S), and volcanic (V). However, some of these 8 classes overlap consistently in the feature space. As a consequence, we exploited the combined use of overlapping aerosol types. Therefore, we merged the types that tend to reflect the same aerosol characteristics, and hence, we evaluate the corresponding effects on the prediction rate of the algorithm. Two pathways were followed, first, the smoke and the polluted continental categories were grouped into the more generic type of small, absorbing particles, and,

second, all the dust-like aerosol types were merged. The different grouping categories are summarized in Table 3.

### 3.2.2 Classifying parameters selection

Next, we performed a sensitivity analysis to identify which classifying properties provide the adequate information to better predict the correct aerosol class. We used three aerosol intensive properties due to the lack of particle linear depolarization ratio profiles to evaluate the strength of the selected classifiers to discriminate among the predefined classes. Two statistical

parameters are used: the total and the partial Wilks' lambda ($\Lambda$; Wilks, 1963) that are widely used e.g., Burton et al. (2012) and Russell et al. (2014). The total $\Lambda$ statistic shows the tendency of the above set of pre-specified classes (or any subset of it) to separate. The partial $\Lambda$ is calculated for each of the intensive properties separately and indicates the discriminatory power of the used intensive property. For both parameters, values range from 0–1. Values near zero show high discriminatory power while values near one show low discriminatory power.

The lowest total $\Lambda$ was found to be 0.033 for the set $\kappa_\beta(355, 1064)$, $S_{aer}^{532}$, and $S_{aer}^{532}/S_{aer}^{355}$; whereas the partial $\Lambda$ is 0.51 for $S_{aer}^{532}/S_{aer}^{355}$, 0.17 for $\kappa_\beta$, and 0.30 for $S_{aer}^{532}$. For this dataset, the low $\Lambda$ value for $\kappa_\beta$ indicates that this variable has the most weight in the classification. The decision for the selected parameters stems solely to the lowest arithmetic value of the total $\Lambda$. Therefore, for the other pairs of parameters the total $\Lambda$ is equally low, ∼0.05, which indicates that, also, a $2\beta + 2\alpha$ lidar setup could be equally used when the algorithm is trained with $\kappa_\beta(355, 532)$. With reference to the lidar ratio, the $S_{aer}^{532}$ and $S_{aer}^{355}$ can

be used interchangeably due to the almost equal total $\Lambda$ (i.e., 0.034).

For the rest aerosol groups reported in Table 2 the total and partial (for $\kappa_\beta$) $\Lambda$ are 0.036 and 0.18 respectively ($7^a$ classes), 0.041 and 0.18 ($7^b$ classes), 0.044 and 0.18 (6 classes), 0.057 and 0.20 (5 classes), and 0.070 and 0.21 (4 classes). The $\Lambda$ shows good discriminatory power for each of the grouping classes, although there is a slight increase in the values as the





number of classes is reduced. This behavior can be ascribed to the high variance of the combined aerosol types which makes the classification less selective.

Figure 5 shows the characteristics of the reference dataset in terms of the lidar ratio and Ångström exponent for the 8 and
4 aerosol classes that represent the maximum and minimum used aerosol groupings. The coloring corresponds to the various
classes and the crosshairs indicate the standard deviation of each of the aerosol layers. The 90 %-confidence ellipses are calculated using the eigenvalues and eigenvectors of the covariance matrix and define the region that contains the 90 % of all the points that can be drawn from the underlying normal class distribution. The various aerosol classes tend to populate specific areas of the graph whereas the overlap of the neighbouring classes is significant, although the classes are better pinpointed as long as we merge classes with similar characteristics. The latter does not reflect the obtained values of the statistical parameters
(total $\Lambda$ increased from 0.033 to 0.070 for 8 classes and 4 classes respectively), however, and as explained above, the reference dataset very well delineates the aerosol types and by combining the neighbouring types the variance increases.

### 3.2.3   Validation of the classifier

In order to evaluate the predictive accuracy of the automatic method it is needed to split the initial reference dataset into a training and a validation dataset. Like this, we use the training dataset to calculate the classification functions and then
submit each observation in the validation dataset to the classification function obtained from the training dataset. For this study, we make use of the leave-one-out cross validation (LOOCV) procedure, also referred to as holdout procedure or simply cross validation, which is a degenerate case of the $k$-fold cross validation, where $k$ is chosen as the total number of samples (Rencher, 2002). The choice of the procedure, even though computational expensive, is used when datasets are sparse and trains the algorithm with as many observations as possible. Each measurement separately is removed from the training dataset
in order to compute the classification rule, and this rule is used to classify the removed observation. The error rate is estimated as a percentage of errors made over the whole set of observations used for testing. For the classification options of the Table 3, the error rate, expectedly, decreases with decreasing number of aerosol classes (39 % for 8 classes, 36 % for $7^a$ classes, 30 % for $7^b$ classes, 28 % for 6 classes, 19 % for 5 classes, and 10 % for 4 classes).

### 3.2.4   Algorithm training including particle depolarization ratio

Several studies have shown the unique information provided by depolarization measurements (e.g., Liu et al., 2008; Tesche et al., 2013; Burton et al., 2014), thus making this intensive property a robust means to discriminate the various aerosol types. Valuable typing information can also be obtained by the color ratio of the particle depolarization ratios when more depolarization channels exist (Burton et al., 2014). As already stated in Sect. 2, the majority of the stations perform depolarization measurements, and profiles are routinely delivered by SCC. However, the reference dataset does not contain depolarization
information because it has been released before the assessment of the quality assurance procedures within EARLINET. Therefore, a method applicable to EARLINET data collected since 2000 is proposed in this work. We investigate the effect of adding depolarization information to the described method as the next releases of EARLINET dataset will contain quality assured particle depolarization profiles and can be used for more accurate aerosol typing. To complement the reference dataset in this



context, we used general literature values for particle linear depolarization ratio at 532 nm (Table 4) in order to train the algorithm. For the clean continental type, the values ingested in the algorithm are retrieved from Burton et al. (2013) and refer to the polluted marine category. The decision for this inconsistency stems to the shortage of clean continental particle depolarization values in literature, however the reported values coincide with the type characteristics described in Sect. 2.2 and the values

used in the CALIPSO typing scheme (Omar et al., 2009).

In this case, the particle linear depolarization ratio replaced the worst performing classifier – i.e., the $S_{aer}^{532}/S_{aer}^{355}$. Hence, the three classifying parameters are the $\delta_{aer}^{532}$, $S_{aer}^{532}$, and $\kappa_\beta(355, 1064)$. Values within the aerosol type range were randomly assigned to each sample and the $\Lambda$ distribution was calculated. Total $\Lambda$ is 0.008. The value of partial $\Lambda$ for $\kappa_\beta$, $S_{aer}^{532}$, and $\delta_{aer}^{532}$ are 0.69, 0.36, and 0.12 respectively. The values found for the partial $\Lambda$ confirm the $\delta_{aer}^{532}$ as the most important classifier for

the considered dataset. For the rest aerosol groups the total and partial (for depolarization) $\Lambda$ are 0.009 and 0.13 respectively ($7^a$ classes), 0.010 and 0.12 ($7^b$ classes), 0.012 and 0.13 (6 classes), 0.072 and 0.67 (5 classes), and 0.090 and 0.69 (4 classes).

For the sake of completeness, the LOOCV method was also performed and the error rate was calculated. Figure 6 presents comparatively the training of the algorithm when depolarization information is available and when not in terms of the total, and partial $\Lambda$ and the error rate of the LOOCV method. The figure highlights the strength of polarization sensitive observations,

while for the 5 and 4 classes (Fig. 6b) the particle linear depolarization ratio becomes less important (in this case the highest weight in the classification corresponds to the lidar ratio) due to the fact that only one dust type represents volcanic and other dust mixtures.

## 4  Results

### 4.1  Testing phase

As a next step, an assessment of the predictive performance of the pre-trained algorithm is made by using a testing dataset. For this, EARLINET data collected during the ACTRIS Summer 2012 intensive measurements (Sicard et al., 2015; Granados-Muñoz et al., 2016b) were chosen to test the automatic typing algorithm. The measurements took place in the period of 8 June–17 July 2012 and were dedicated to Saharan dust studies and also featured two field campaigns such as PEGASOS (Pan European Gas Aerosols Interaction Study) and Charmex (Chemistry Aerosol Mediterranean Experiment). During that period,

157 measurements were performed, out of which 42 measurements delivered 3 backscatter and 2 extinction coefficient profiles. The description of aerosol type distribution over Europe during the campaign was obtained following the procedure shown in Sect. 2.1 (Papagiannopoulos et al., 2016b). The testing dataset comprises of 47 samples, 21 of which yield depolarization ratio values. Table 5 provides the mean values of the intensive parameters for each available category in accordance with Table 2.

### 4.2  Application of the methodology to EARLINET data - case studies

To showcase the steps of the automatic classification, we apply it to two selected cases for the 8 classes and for the classifying parameters: $S_{aer}^{532}/S_{aer}^{355}$, $S_{aer}^{532}$, and $\kappa_\beta(355, 1064)$. For the case in Sect. 2.1, the automatic algorithm labelled the aerosol layer





as pure dust, $D_M = 1.0$ and the normalized probability 62 %. This coincides with our findings and highlights the strength of the classification, albeit this example corresponds to a pure aerosol layer with no level of mixing with other aerosol types.

The second case refers to a more complicated aerosol scene. The Athens EARLINET station (Fig. 7) on 22 May, 2014 observed an aerosol layer mostly in the height range between 1.5 and 3 km (Papayannis et al., 2016). Within this layer the mean value of backscatter (extinction) related Ångström exponent is $0.9 \pm 0.1$ ($1.0 \pm 0.4$). The lidar ratio presents mean values in the layer $40 \pm 7$ sr and $39 \pm 6$ sr at 355 nm and 532 nm, respectively. The color ratio of the lidar ratios shows a wavelength independent layer with values $1.1 \pm 0.2$. The retrieved error corresponds to the standard deviation of the retrieved quantity calculated within the layer.

In the following, a 6-day FLEXPART backward trajectory indicates the pattern of the origin of airmasses. Figure 8 shows the total column sensitivity of the particles found over the station in between 1.5–3 km, it highlights the motion of the particles in a north-easterly direction towards the Aral Sea and Kazakhstan. This area is an active dust source due to the extreme desiccation of the lake (Ginoux et al., 2012). Therefore, the path of the air masses arriving over Athens suggests a mixture of dust, marine and biomass burning particles, originating from the arid areas of the Aral Sea, as well as the agricultural fires in former Soviet Union countries, enriched with marine particles during their overpass over the Black Sea (Papayannis et al., 2016). The automatic algorithm classified the layer as mixed dust, $D_M = 2.5$ and normalized probability 32 %, and the second closest class was clean continental, $D_M = 3$ and normalized probability 23 %. Although the class with the minimum estimated distance agrees with our investigation, the inferred type will not be taken into account. The very low probability indicates that more than one distance is beyond the accepted threshold, therefore the classes are almost equidistant. This demonstrates that the manual typing procedure can better type the aerosol layer, but also that adopted fixed thresholds are conservative, i.e., type assignment is not possible for ambiguous scenes.

## 4.3 Comparing the automatic classification with manual analyzed data

The performance of the algorithm with respect to the testing dataset is presented. For each of the grouping classes, as those listed in Table 3, the confusion matrices have been calculated (not shown) and the accuracy of the model is presented, here, alongside with the recall ($R$) and precision ($P$). The confusion matrix describes the performance of the classifier on a testing dataset for which the typing is already known. Accuracy shows how many times the predicted aerosol type agrees with the true aerosol type and is the ratio of the correctly classified instances (true positives) to the total number of instances. Recall of each aerosol group is defined as the number of true positives ($T_p$) over the number of true positives plus the number of false negatives ($F_n$). The false negatives represent the instances of cases when we predicted a different aerosol type than the true one.

$$R = \frac{T_p}{T_p + F_n} \tag{4}$$





Precision of each aerosol group is defined as the number of true positives ($T_P$) over the number of true positives plus the number of false positives ($F_p$). The false positives represent the number of instances when we predicted a specific aerosol type although different from the true one.

$$P = \frac{T_p}{T_p + F_p} \tag{5}$$

For an ideal classification procedure both $F_p = F_n = 0$ and so $R = P = 1$. For real classification systems false positive and/or false negatives may occur and consequently values of $R$ and $P$ between 0 and 1 are possible. The more the value of $R$ and $P$ is far from 1 the less the classification system behaves like an ideal one. In particular, a value of $R$ ($P$) close to 1 indicates the generation of negligible number of false negatives (positives) during the classification procedure.

In Fig. 9, the bar plot shows comparatively the predictive accuracy of the algorithm when compared to manual analyzed data for the different aerosol classes in both the cases in which the depolarization information is available (in orange) or not (in brown). Without depolarization ratio information, the accuracy of the model increases with decreasing number of classes. The lowest value was obtained for 8 classes (59 %) and the highest for 4 classes (90 %). With depolarization ratio information, the accuracy for 8 classes equals to 69 % and exceeds the 80 % for the rest aerosol classes. When comparing the accuracy of the model with and without depolarization ratio, it appears to be significantly higher until 6 classes where, further, the discrepancy diminishes ($\sim$10 %). In general, it becomes evident that the particle linear depolarization ratio increases the ability for predicting correctly the aerosol type. Given the high accuracy, a $3\beta + 2\alpha$ configuration showed that 6 aerosol classes, as well as 5 and 4, can provide a robust classification. Instead, the training of the classification with depolarization measurements enhances the predictability strength and can provide finer aerosol classification (for 8 classes, accuracy $\sim$70 %).

Table 6 summarizes the results when using as classifying parameters: $S_{aer}^{532}/S_{aer}^{355}$, $S_{aer}^{532}$, and $\kappa_\beta(355, 1064)$, with respect to recall and precision and offer a better insight in the performance of each aerosol type. Next to the number of classes, between parentheses, the number of aerosol layers that passed the screening criteria as those described in Sect. 3.1 is provided. It is, thus, worth noting that the numbers increase when the aerosol types are combined. The mixed marine and clean continental aerosol types yield high recall and precision (values>80 %) throughout the different aerosol classes, highlighting the ability of the classifier to correctly label them. The aerosol types that performed worse are the smoke and polluted continental aerosol types due to the similarities in the intensive optical properties. However, when combining them into a single aerosol class (see $7^b$, 6, and 4 classes) precision and recall increase significantly. Given the noticeable signature of dust particles precision is high, whereas the recall is 30 % and this can be assigned to the lack of depolarization measurements. Similarly, recall increases as soon as volcanic, mixed and polluted dust are included in the same all-dust category (see $7^a$, 6, 5, and 4 classes). Note that mixed dust and polluted dust aerosol types are not reported in the tables due to the fact that they are not present in Table 5 and these parameters cannot be evaluated.

Table 7 is similar to Table 6 and reports the recall and precision when depolarization information is available. Clean continental aerosol, again, yields high recall and precision for all the different aerosol groups. Polluted continental performed the worse and, expectedly, showed the same behavior as before when compared with smoke into a single type. Alternately, dust





is precisely identified for all the aerosol classes. This result indicates that depolarization measurements facilitate the correct dust typing. It is noteworthy, that although the findings are promising the test dataset is limited and do not cover all the aerosol classes.

## 5   Summary and Conclusions

The characterization of the vertical aerosol distribution is needed for accurate radiative-transfer modeling. Automatic procedures to classify aerosols objectively and within near-real-time scales are employed. An automatic classification procedure based only on EARLINET data was presented. Here, we modified an automatic algorithm to satisfy the network's requirements and needs. A Wilks' lambda analysis was performed on EARLINET data and the three best performing classifying parameters were: the lidar ratio at 532 nm; the color ratio of the lidar ratios at 355 nm and 532 nm and the backscatter-related 355-to-

1064- nm Ångström exponent. Nevertheless, the other intensive parameters using the available wavelengths can be equally used as the analysis showed similar values. Furthermore, the number of aerosol classes has been investigated for a maximum of 8 and minimum 4. Prior to evaluating the performance of the algorithm, the leave-one-out cross validation procedure was performed on the reference dataset and the error rate decreased monotonously from 39 % to 10 % with decreasing number of aerosol classes. The prediction of the automatic classification showed positive results when compared against already classified

EARLINET data. In particular, the positive learning success for 8 (59 %), 7 (69 % for $7^a$ and $7^b$ classes), 6 (76 %), 5 (76 %) and 4 (90 %) classes indicates that the fewer aerosol classes (6, 5, and 4 classes) provide a confident but, nonetheless, a coarser classification. To be more precise, the high accuracy (76 %) coupled with the low error rate of the cross-validation (28 %) for 6 classes offers a good starting point for a classification with a $3\beta + 2\alpha$ lidar configuration.

Besides, the training of the algorithm with literature depolarization ratio values decreased the error rate of the leave-one-

out cross validation from 30 % to 7 % and increased the predictive accuracy. For 8 classes the predictive accuracy reached 69 % while for the rest it remained well above 80 % (for $7^a$: 86 %, for $7^b$: 83 %, for 6: 93 %, for 5: 86 %, and for 4: 93 %). Therefore, this finding suggests that the algorithm in this case can be used for finer aerosol classification and also delineates the discriminatory power of depolarization ratio. Specifically, 7 aerosol classes (either D+V, MD, PD, CC, MM, PC, S or D, V, MD, PD, CC, MM, PC+S) seem to be adequate to provide reasonable typing results. However, the obtained results refer to

a small testing dataset that consists of pure aerosol types and underestimates the aerosol mixtures of the classification.

The presented automatic algorithm is based only on EARLINET data and is set to accommodate EARLINET measurements covering as much of its measurement record as possible. Specifically, Raman lidar systems with $3\beta + 2\alpha$ and $2\beta + 2\alpha$ configuration with and without particle depolarization ratio can be used for the aerosol classification. The manageability of the algorithm regarding the reference dataset, the number of the aerosol classes and the classifying parameters make the method

easily adaptable and handled by individual users. The training dataset can be easily enlarged with high quality typing data coming from a multitude of EARLINET stations and a longer time record. Moreover, new classifying parameters, particle linear depolarization ratio at more wavelengths and aerosol extinction coefficient in the infrared, can be easily added as the observing capacity increases.



The use of the method network-wide will homogenize and standardize the aerosol typing towards a new EARLINET product. The implementation of the method into the SCC will create a complete automatic lidar analysis, that is, from the retrieval of optical properties to aerosol classification. Further, an intercomparison of the developed method against methods which make use also of aerosol optical property modeling could improve from one side the optimization of aerosol property models and

from the other side the tuning of aerosol types and reference dataset. This method, even if developed on the basis of EARLINET and its variable instrumental capability, can be applied to all of the aerosol lidar systems as those part of GALION as well as to future lidar-based satellite missions (e.g., the Earth Cloud Aerosol and Radiation Explorer (EarthCARE) satellite mission). In future, a combination of the few sophisticated EARLINET-type lidars and extended networks of automated single-wavelengths backscatter lidars (as ceilometers, Wiegner et al., 2014) might be beneficial with aerosol typing provided at "anchor stations",

and the spatial extent of the layers can be provided by the continuous observations of the ceilometers. This will also offer a unique data set for evaluation of models.

*Acknowledgements.* The financial support for EARLINET in the ACTRIS Research Infrastructure Project by the European Union's Horizon 2020 research and innovation programme under grant agreement no. 654169 in the Seventh Framework Programme (FP7/2007–2013) is gratefully acknowledged. The research leading to these results has received funding from European Union's Horizon 2020 research and in-

novation programme under grant agreement no. 602014 (project ECARS (East European Centre for Atmospheric Remote Sensing)) and from European Union's Horizon 2020 research programme for Societal challenges - smart, green and integrated transport under grant agreement no. 723986 (project EUNADICS-AV (European Natural Disaster Coordination and Information System for Aviation)).



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



**Table 1.** Number of classified aerosol layers adapted from Schwarz (2016). The mixtures category is comprised of two or more (values in parentheses) pure aerosol types.

| Aerosol type | All analyzed | Only from $3\beta + 2\alpha$ |
|---|---|---|
| Clean Continental (CC) | 45 | 5 |
| Polluted Continental (PC) | 95 | 19 |
| Dust (D) | 41 | 6 |
| Mixed Dust (MD) | 56 | 9 |
| Polluted Dust (PD) | 14 | 3 |
| Smoke (S) | 24 | 7 |
| Volcanic (V) | 21 | 4 |
| Mixtures | 348 (170) | 35 (11) |
| Total | 644 | 88 |



**Table 2.** Reference dataset: mean type-dependent intensive properties along with the standard deviation.

| Type | $\kappa_\beta(355,1064)$ | $\kappa_\beta(532,1064)$ | $\kappa_\beta(355,532)$ | $\kappa_\alpha(355,532)$ | $S_{aer}^{355}$ [ sr] | $S_{aer}^{532}$ [ sr] | # Samples |
|------|------|------|------|------|------|------|------|
| CC | $1.0 \pm 0.2$ | $1.0 \pm 0.3$ | $1.3 \pm 0.3$ | $1.7 \pm 0.6$ | $50 \pm 8$ | $41 \pm 6$ | 9 |
| PC | $1.3 \pm 0.3$ | $1.3 \pm 0.2$ | $1.4 \pm 0.6$ | $1.7 \pm 0.5$ | $69 \pm 12$ | $63 \pm 13$ | 16 |
| D | $0.4 \pm 0.1$ | $0.4 \pm 0.1$ | $0.3 \pm 0.2$ | $0.3 \pm 0.4$ | $58 \pm 12$ | $55 \pm 7$ | 9 |
| MD | $0.5 \pm 0.2$ | $0.4 \pm 0.3$ | $0.7 \pm 0.3$ | $0.5 \pm 0.3$ | $42 \pm 4$ | $47 \pm 6$ | 10 |
| PD | $0.9 \pm 0.3$ | $0.8 \pm 0.1$ | $1.0 \pm 0.5$ | $0.6 \pm 0.2$ | $54 \pm 8$ | $64 \pm 9$ | 5 |
| MM | $0.8 \pm 0.1$ | $0.8 \pm 0.2$ | $1.0 \pm 0.3$ | $0.9 \pm 0.3$ | $25 \pm 7$ | $24 \pm 8$ | 8 |
| S | $1.3 \pm 0.1$ | $1.3 \pm 0.1$ | $1.2 \pm 0.3$ | $1.3 \pm 0.3$ | $81 \pm 16$ | $78 \pm 11$ | 7 |
| V | $0.1 \pm 0.1$ | $0.4 \pm 0.3$ | $0.2 \pm 0.3$ | $0.2 \pm 0.3$ | $50 \pm 11$ | $48 \pm 13$ | 5 |





**Table 3.** Aerosol types that constitute the classes investigated. CC stands for Clean Continental, PC stands for polluted continental, D stands for dust, MD stands for mixed dust, PD stands for polluted dust, MM stands for mixed marine, S stands for smoke, and V stands for volcanic particles.

| # Types | Groups of aerosol types | | | | | | |
|---|---|---|---|---|---|---|---|
| 8 | D | V | MD | PD | CC | MM | PC | S |
| 7[a] | D+V | MD | PD | CC | MM | PC | S |
| 7[b] | D | V | MD | PD | CC | MM | PC+S |
| 6 | D+V | MD | PD | CC | MM | PC+S |
| 5 | D+V+MD+PD | CC | MM | PC | S |
| 4 | D+V+MD+PD | CC | MM | PC+S |



**Table 4.** The mean and standard deviation of the particle depolarization ratio used for the pre-specified classes and the corresponding bibliographic references.

| Type | $\delta_{aer}^{532}$ | References |
|---|---|---|
| Clean Continental | $0.04 \pm 0.02$ | Burton et al. (2013) |
| Polluted Continental | $0.05 \pm 0.03$ | Burton et al. (2013) |
| Dust | $0.30 \pm 0.01$ | Groß et al. (2011) |
| Mixed Dust | $0.15 \pm 0.02$ | Groß et al. (2016) |
| Polluted Dust | $0.20 \pm 0.05$ | Burton et al. (2013) |
| Marine | $0.03 \pm 0.01$ | Groß et al. (2013) |
| Smoke | $0.10 \pm 0.04$ | Burton et al. (2013) |
| Volcanic | $0.33 \pm 0.03$ | Pappalardo et al. (2013) |



**Table 5.** Testing dataset: mean type-dependent intensive properties along with the standard deviation.

| Type | $\kappa_\beta(355,1064)$ | $\kappa_\beta(532,1064)$ | $\kappa_\beta(355,532)$ | $\kappa_\alpha(355,532)$ | $S_{aer}^{355}$ [ sr] | $S_{aer}^{532}$ [ sr] | # Samples |
|------|------|------|------|------|------|------|------|
| CC | $1.2 \pm 0.4$ | $1.6 \pm 0.4$ | $1.3 \pm 0.2$ | $1.6 \pm 0.4$ | $43 \pm 5$ | $38 \pm 6$ | 12 |
| PC | $1.3 \pm 0.4$ | $1.4 \pm 0.3$ | $1.3 \pm 0.3$ | $1.2 \pm 0.3$ | $52 \pm 6$ | $56 \pm 8$ | 8 |
| D | $0.3 \pm 0.3$ | $0.2 \pm 0.3$ | $0.3 \pm 0.2$ | $0.0 \pm 0.2$ | $54 \pm 11$ | $54 \pm 9$ | 13 |
| MM | $0.8 \pm 0.2$ | $1.2 \pm 0.5$ | $0.9 \pm 0.3$ | $0.9 \pm 0.3$ | $27 \pm 9$ | $24 \pm 8$ | 8 |
| S | $1.6 \pm 0.4$ | $1.6 \pm 0.5$ | $1.5 \pm 0.3$ | $1.6 \pm 0.4$ | $54 \pm 9$ | $61 \pm 6$ | 6 |



**Table 6.** Recall ($R$), and precision ($P$) estimated from the classification matrices for 8, $7^a$, $7^b$, 6, 5, and 4 classes. The values between parentheses correspond to the number of layers passed the screening criteria.

| Types | $R$ [%] | $P$ [%] | Types | $R$ [%] | $P$ [%] | Types | $R$ [%] | $P$ [%] |
|---|---|---|---|---|---|---|---|---|
| **8 classes (29/47)** | | | **$7^a$ classes (29/47)** | | | **$7^b$ classes (34/47)** | | |
| CC | 100 | 82 | CC | 100 | 89 | CC | 100 | 90 |
| D | 30 | 100 | D+V | 55 | 100 | D | 27 | 100 |
| MM | 100 | 100 | MM | 100 | 100 | MM | 100 | 100 |
| PC | 25 | 33 | PC | 50 | 50 | PC+C | 78 | 100 |
| S | 0 | - | S | 0 | - | | | |
| **6 classes (33/47)** | | | **5 classes (35/47)** | | | **4 classes (39/47)** | | |
| CC | 100 | 80 | CC | 100 | 82 | CC | 100 | 85 |
| D+V | 58 | 100 | D+V+PD+MD | 100 | 87 | D+V+PD+MD | 100 | 87 |
| MM | 100 | 100 | MM | 100 | 100 | MM | 100 | 100 |
| PC+S | 63 | 100 | PC | 20 | 33 | PC+S | 56 | 100 |
| | | | CC | 0 | - | | | |





**Table 7.** Recall ($R$) and precision ($P$) estimated from the classification matrices for 8, $7^a$, $7^b$, 6, 5, and 4 classes when particle linear depolarization ratio measurements are available. The values between parentheses correspond to the number of layers passed the screening criteria.

| Types | $R$ [%] | $P$ [%] | Types | $R$ [%] | $P$ [%] | Types | $R$ [%] | $P$ [%] |
|---|---|---|---|---|---|---|---|---|
| **8 classes (13/21)** | | | **$7^a$ classes (14/21)** | | | **$7^b$ classes (12/21)** | | |
| CC | 100 | 80 | CC | 100 | 100 | CC | 100 | 100 |
| D | 67 | 100 | D+V | 80 | 100 | D | 67 | 100 |
| MM | - | - | MM | - | - | MM | - | - |
| PC | 50 | 100 | PC | 100 | 50 | PC+S | 100 | 100 |
| S | 0 | - | S | 0 | - | | | |
| **6 classes (14/21)** | | | **5 classes (14/21)** | | | **4 classes (14/21)** | | |
| CC | 100 | 100 | CC | 100 | 100 | CC | 100 | 100 |
| D+V | 88 | 100 | D+V+PD+MD | 100 | 89 | D+V+PD+MD | 100 | 89 |
| MM | - | - | MM | - | - | MM | - | - |
| PC+S | 100 | 100 | PC | 50 | 50 | PC+S | 67 | 100 |
| | | | S | 0 | - | | | |



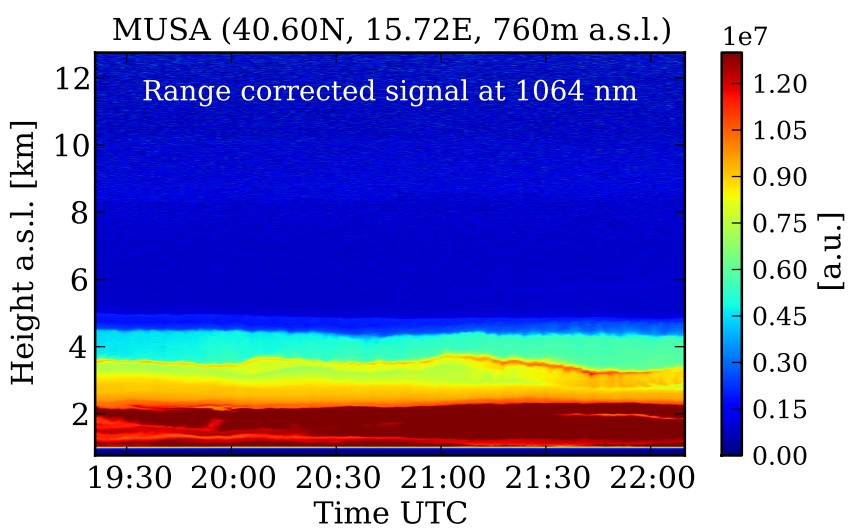

**Figure 1.** Temporal evolution of the 1064-nm range corrected lidar signal obtained with the MUSA system in Potenza on 14/07/2011, 19:20–22:10 UTC.



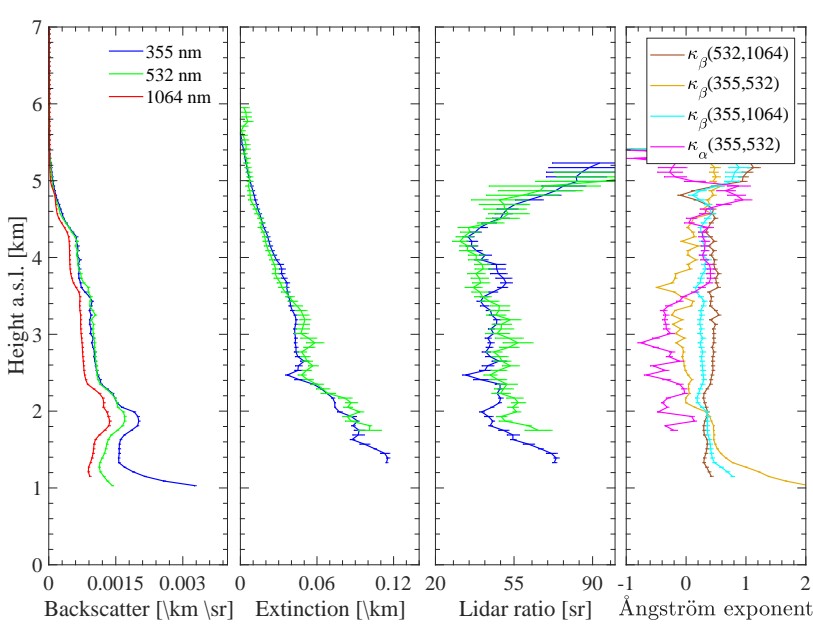

**Figure 2.** Optical profiles measured in Potenza, on 14 July 2011, 19:20–22:10 UTC with a multiwavelength Raman lidar. The error bars correspond to the standard deviation.



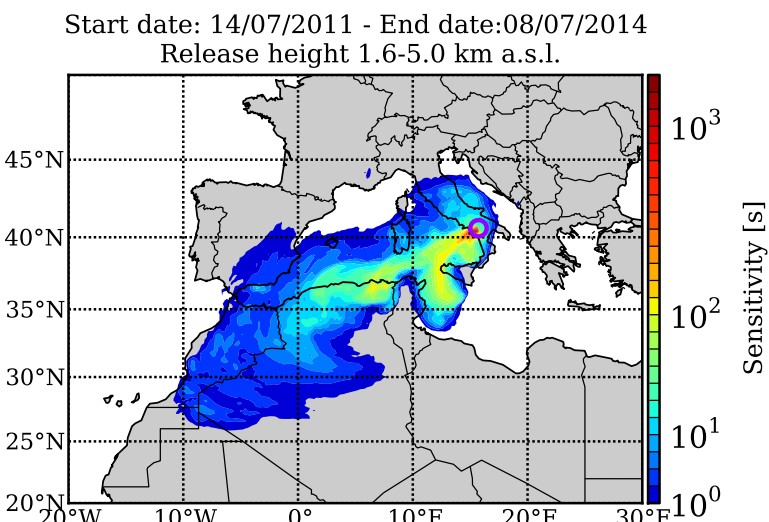

**Figure 3.** FLEXPART footprint for the airmass travelling below 2 km height and arriving at Potenza between 1.5–5 km at 22:00 UTC on 14 July 2011. The colors are coded with respect to the logarithm of the integrated residence time in a grid box in seconds for a 5-day integration time.





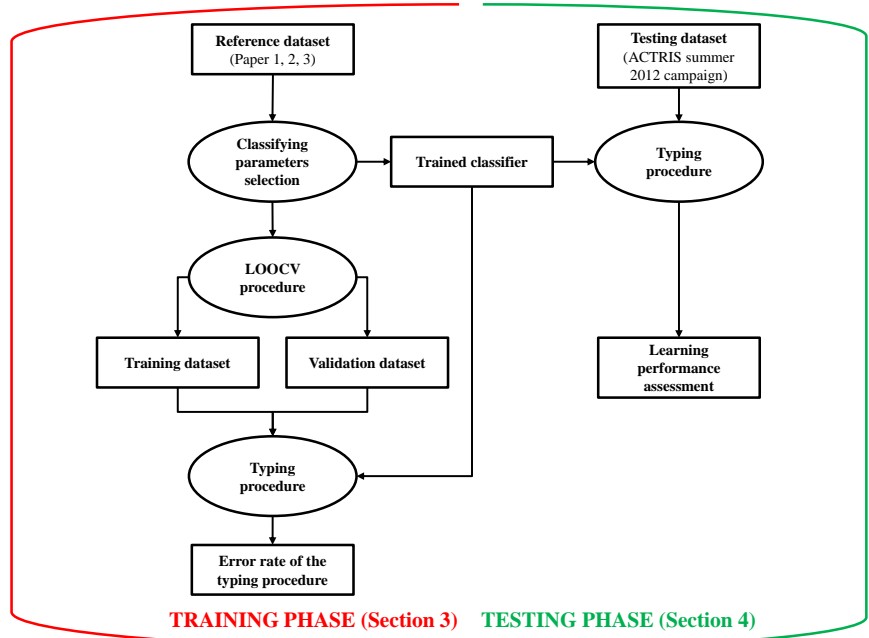

**Figure 4.** Flowchart of the methodology. First, well characterized aerosol layers are grouped into meaningful classes that represent the reference dataset: Paper 1 (Papagiannopoulos et al., 2016a), Paper 2 (Pappalardo et al., 2013), Paper 3 (Schwarz, 2016). Second, an analysis is performed to determine the best performing classifying parameters among the available intensive parameters. Third, based on the reference dataset the selected classifier is validated using the Leave-one-out cross validation (LOOCV) procedure in order to ensure correct aerosol type separation. Finally, the trained typing algorithm is applied to an independent and manually typed dataset (the testing dataset) for the assessment of the algorithm performance. Note that both phases have been applied with and without the depolarization ratio.



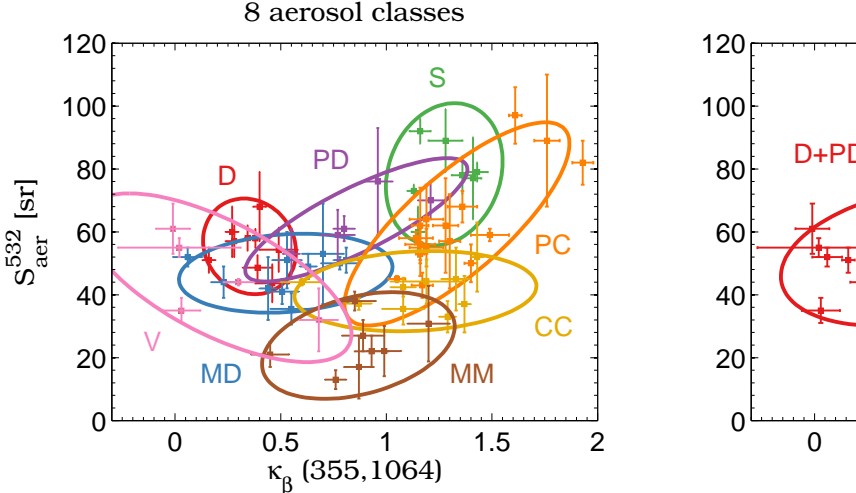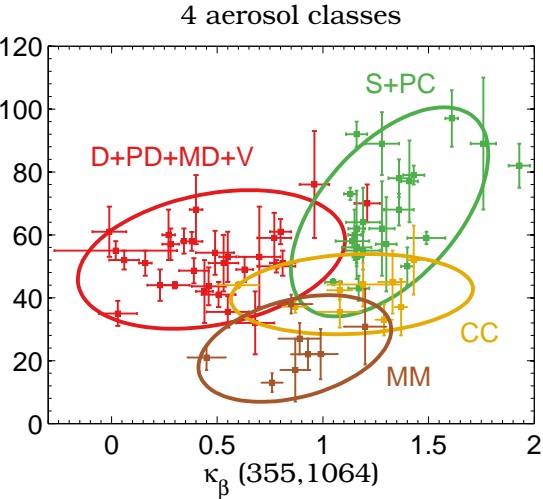

**Figure 5.** Colored pre-specified classes and 90 % confidence ellipses for 8 and 4 aerosol classes. The error bars correspond to the standard deviation of the selected mean intensive properties. CC stands for Clean Continental, D stands for dust, MD stands for mixed dust, MM stands for mixed marine, PD stands for polluted dust, PC stands for polluted continental, S stands for smoke, and V stands for volcanic particles.



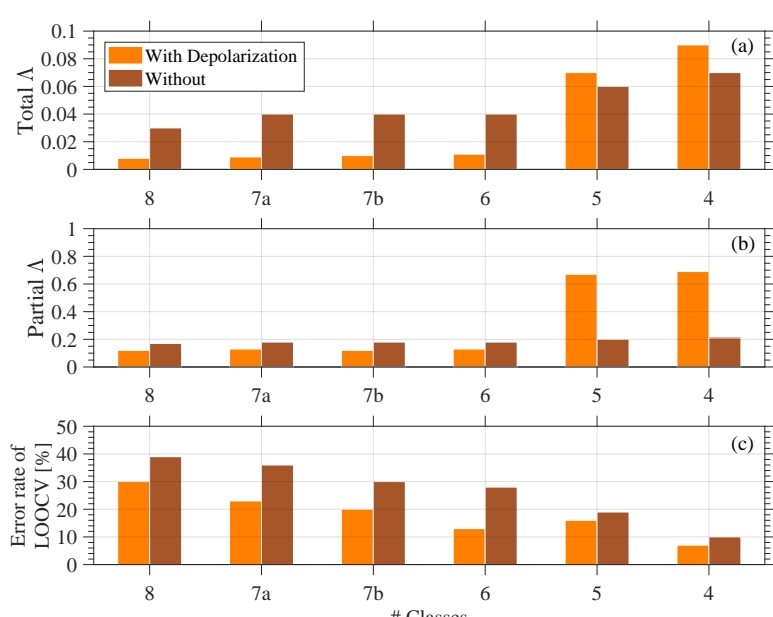

**Figure 6.** Bar plots showing a) the total $\Lambda$, b) the partial $\Lambda$, and c)error rate of LOOCV when comparing the training of the algorithm with (i.e., $S_{aer}^{532}/S_{aer}^{355}$, $S_{aer}^{532}$, and $\kappa_{\beta}$) and without (i.e., $\delta_{aer}^{532}$, $S_{aer}^{532}$, and $\kappa_{\beta}$) particle linear depolarization values. For the partial $\Lambda$, the brown bars correspond to the backscatter-related Ångström exponent and orange one to the particle linear depolarization ratio because because they represent the most significant classifying parameter of the classification.





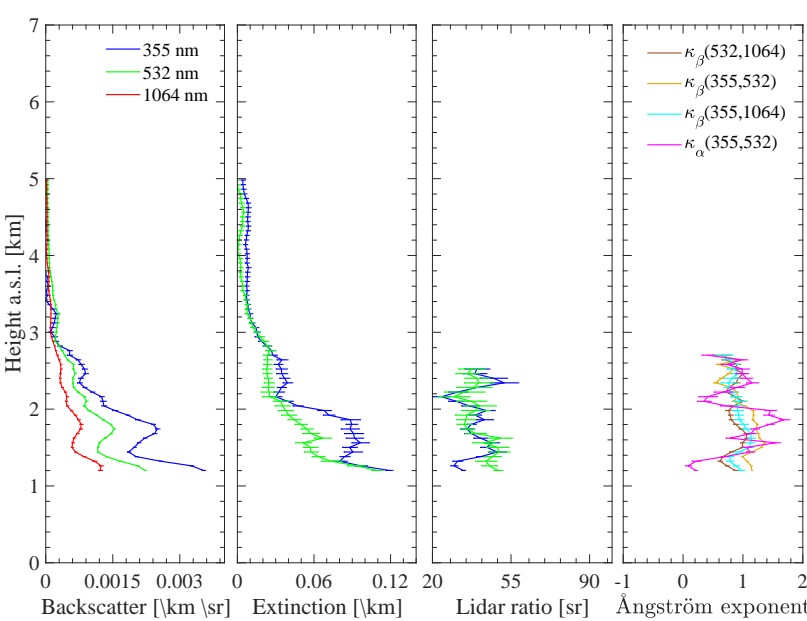

**Figure 7.** Optical profiles measured at Athens, on 22 May 2014, 20:28–21:28 UTC with a multiwavelength Raman lidar. The error bars correspond to the standard deviation.





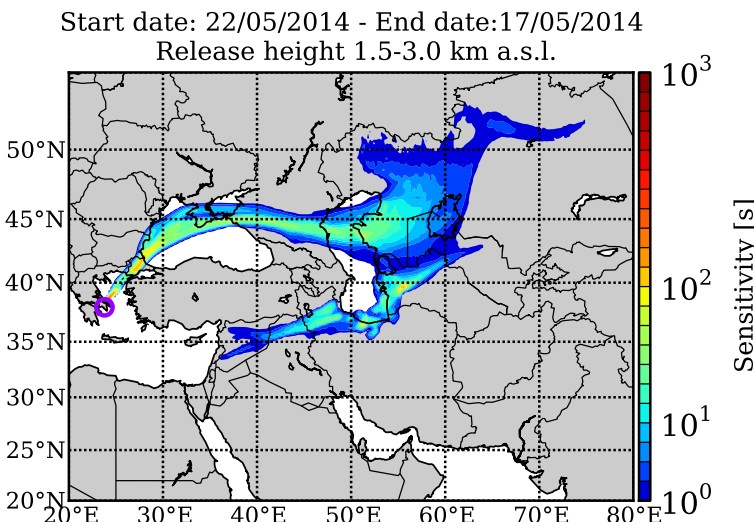

**Figure 8.** FLEXPART footprint for the air mass travelling below 2 km height and arriving over Athens between 1.5 and 3.0 km a.s.l. at 20:28–21:28 UTC on 22 May 2014. The colors represent the logarithm of the integrated residence time in a grid box in seconds for 6-day integration time.





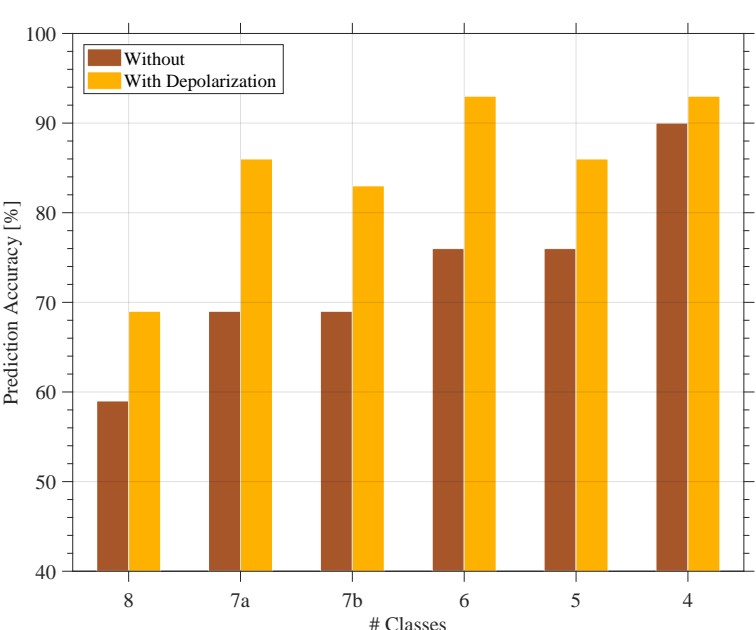

**Figure 9.** Prediction accuracy for the different aerosol classes and with/without depolarization information.