# Peer review of "An automatic observation-based aerosol typing method for EARLINET"

_Atmospheric Chemistry and Physics, 2018_

## Referee Comment (RC1) · S. P. Burton (Referee) · 28 Jun 2018

General:

This paper describes the adaptation of a flexible automated typing algorithm to EAR-LINET network lidar data. It's great to see this product being produced for such a large and continuing dataset. As the authors point out, aerosol typing is potentially useful for developing a better understanding of aerosol sources and for improving the accuracy of satellite retrievals and climate and weather models. Producing typing data for EARLINET makes some of these goals become more accessible. The scientific methodology is good and the analysis and testing are thorough and include the introduction of useful new statistics and tools. The success rate of the algorithm is not

always high, but the authors present an analysis of how this depends on the observed variables and how the algorithm would be adapted for making use of additional variables. The precision of the language could be improved, and I have a few suggestions about this and about some other aspects of the analysis below.

Specific comments:

Figures 2 and 7. What are the resolution of the extinction and backscatter profiles? How is the lidar ratio calculated? Were the extinction and backscatter at the same resolution before taking the ratio? I ask because there are differences in the shape of extinction and backscatter in each of the discussed layers that do not seem particularly consistent with the idea that each layer is a specific coherent aerosol type. For example, the "layer" below 2 km in Figure 2 has a completely different shape in backscatter vs. extinction, leading to large variability in the lidar ratio, much larger than the suggested error bars. Do you think this variability is real, or could it be that the local maximum (seen in backscatter) is smoothed out in extinction by a coarser vertical resolution? If it's real, is it likely this is a single consistent aerosol type in this layer? Similarly, the different slopes in the "layer" between 3.5 and 5 km lead to a very large slope in the lidar ratio that does not seem consistent with the idea that this is a single aerosol type. If this variability is spurious, it is liable to create additional apparent noise in the classification that is not really related to aerosol variability within classes. (Of course, spurious error would also be a concern in general, not just for classification.)

Would you say that there is a possibility for error in the determination of "truth" aerosol types? If so, it would be good to see some discussion of that. I also have a particular question about the interpretation of the influence of marine aerosol in the two case studies that are discussed at length. FLEXPART, Figures 3 and 8, seems like a very nice tool for information on aerosol source. Figure 3 seems to show a large fraction of the incidence below 2 km (a lot of the green and yellow) as being in the Mediterranean Sea, but in the discussion, no mention is made of marine influence and the case is described as "pure dust". On the other hand, in Figure 8, an apparently lesser proportion of the trajectories below 2 km are seen over the Black Sea and the Caspian Sea, and this layer is described as a mixture "enriched with marine particles during their overpass over the Black Sea". Can you clarify how we can know that there is marine influence in one case and not the other? It seems that it would be particularly difficult to say definitively when aerosol types are "pure" rather than mixed, using this method. Can you clarify whether there are additional factors that go into these judgements besides the FLEXPART tool and what uncertainties are associated with those judgements?

Section 3.2.4. When you add particle depolarization as a classification variable, I think you should still keep the original three variables. You have already shown that all three variables are sufficiently independent to be useful, so adding an independent fourth variable would be expected to produce the best classification method. I'm not following why you avoid using more than 3 variables.

P4 L14 The idea that the spectral ratio of lidar ratio indicates smoke aging is still a hypothesis, based empirically on a small number of suggestive cases. Describing this relationship as "robust" overstates the case, I think. Not all of the references actually support the statement. For example, Samaras et al. 2015 have no measurements related to smoke age, but rather take it as given, substituting the spectral ratio of lidar ratios as a proxy for smoke age. Please don't use a reference to support a hypothesis that merely made use of the hypothesis (at least not without more explanation). Anyway, the current manuscript doesn't relate to smoke aging. You could easily remove the statement and avoid controversy. At least don't overstate it and please remove references that do not really support the statement.

Should the "clean" in the label "clean continental" be taken literally? The description of the clean continental category is obliquely defined here as a mixture of polluted continental and clean marine aerosol and indeed the data in Figure 5 also seem to support its interpretation as a mixture of the two. I think this is an interesting way to think about this type, perhaps much more useful than the standard way of thinking

of it as a type that, unlike all the others, is defined by an extensive aerosol property (low aerosol loading). Any comment on this? Would a case that has the intensive properties of the clean continental class but a significant amount of aerosol optical depth (so therefore not particularly "clean") be considered "clean continental" in your analysis?

P9-10. The information about Mahalanobis distance is basically repeated from earlier work. You could simplify by referencing Burton et al. 2012 and noting the different thresholds.

P10, L7. Similar probabilities for two different classes do not indicate mixing between those two classes. This only reflects that those two classes are close to each other in your measurement space. For example, any point that is close to your "smoke" class is also close to your "polluted continental" class because the classes are close to each other. If it actually is smoke plus a little bit of marine influence, the 2nd closest class will still be "polluted continental", not marine.

Section 3.2.2 classifying variables selection. I think choosing variables solely based on Wilks' partial lambda may not catch everything. It may be that different variables have more power to separate different subsets of classes. For example, depolarization obviously has a lot of power to separate dust classes and almost no power to separate non-dust classes. If you had a variable that helped separate smoke from polluted continental, even partially, but did nothing else, it may have a poor partial lambda but it would nevertheless be extremely valuable. I suggest a plot similar to figure 10 in Burton et al. 2012 as a way to understand more thoroughly what each variable contributes to separating the classes. By looking at the variability of each variable within each class you can see where the overlaps are in every dimension. Your figure 5 also does this but it's usefulness maxes out at 2 dimensions, since it is very difficult to visualize more than 2 dimensions in a literal space. Your Wilks' lambda analysis suggests that the ratio of lidar ratios has significant discriminatory power, but Figure 5 does not reveal how (that is, whether some sets of classes that look like they overlap might be distinguished by

the 355 nm lidar ratio or the spectral lidar ratio). It would be nice to have a visualization that answers that question. This would also address the question of whether spectral lidar ratio separates smoke and pollution aerosol (Muller et al. 2007a) which would be an interesting discussion in itself.

P16, Perhaps you could discuss more explicitly the tradeoff between more classes and less classes. All of your statistics (except Wilks' lambda) seem to show better performance with fewer classes, but that could be taken to an extreme. That is, with only one class, there would be no errors at all! How do you address this tradeoff?

Do you plan to share these aerosol typing results publically? What about the training database of manually typed samples? Also please include links to the EARLINET database.

Typos and requests for clarification:

P2, last sentence: Both cluster analysis techniques and supervised classification techniques need the number of groups as input.

P4 L7: "operates" should be "operate"

P5, L18: talks about the region of incomplete overlap. What altitude does this go up to?

P5, L22-23: the sentence "the aforementioned layers" is unclear and should be reworded. I think you are suggesting that the intensive properties are approximately constant throughout each layer, but that does not really appear to be true, so perhaps I'm misunderstanding the wording.

P5, L24 and throughout: there are four Angstrom exponents discussed but often the text refers to "Angstrom exponent" as if there is only one. Please clarify which one you mean. Likewise, it should be specified which wavelength is meant when "lidar ratio" is used.

P6 L23. I recommend taking more care about using the word "absorbing". It seems that "more absorbing" is here used as a synonym for "higher lidar ratio", but lidar ratio depends on particle size and other factors as well as light absorption and is really not as direct an indicator as this language suggests. Also P11 L4 and L14 (continental pollution is not necessarily absorbing); P8 L5 (smoke is often absorbing but not always highly absorbing especially when aged); and perhaps elsewhere.

P7, L14. Delete "mainly". Although the variability in the lidar ratio is a "hot topic" there is also significant variability in, for instance, extinction Angstrom exponent.

P7, L31. The suggestion for CALIPSO to add a dust+marine type was made several times before 2016 also, for example Kim et al. 2013, Burton et al. 2013, Rogers et al. 2014

Kim, M.-H., Kim, S.-W., Yoon, S.-C., and Omar, A. H.: Comparison of aerosol optical depth between CALIOP and MODIS-Aqua for CALIOP aerosol subtypes over the ocean, Journal of Geophysical Research: Atmospheres, 10.1002/2013jd019527, 2013.

Rogers, R. R., Vaughan, M. A., Hostetler, C. A., Burton, S. P., Ferrare, R. A., Young, S. A., Hair, J. W., Obland, M. D., Harper, D. B., Cook, A. L., and Winker, D. M.: Looking Through the Haze: Evaluating the CALIPSO Level 2 Aerosol Layer Optical Depth using Airborne High Spectral Resolution Lidar Data, Atmos. Meas. Tech. , 7, 4317-4340, 10.5194/amt-7-4317-2014, 2014.

P8 L10 and in the references: I think Pereira should be Nepomucino Pereira (that is, the first author appears to have a two-part surname).

P9 first paragraph and third paragraph: There are a few places, including these 2 paragraphs, where the wording is awkward with several errors in English language usage that make them hard to understand. Please reword for clarity.

P12 L28, I think you mean Burton et al. 2015, not 2014. Burton, S. P., Hair, J. W.,

Kahnert, M., Ferrare, R. A., Hostetler, C. A., Cook, A. L., Harper, D. B., Berkoff, T. A., Seaman, S. T., Collins, J. E., Fenn, M. A., and Rogers, R. R.: Observations of the spectral dependence of linear particle depolarization ratio of aerosols using NASA Langley airborne High Spectral Resolution Lidar, Atmos. Chem. Phys., 15, 13453-13473, 10.5194/acp-15-13453-2015, 2015.

P13, L25 & L27, and throughout. When you count measurements or samples, what defines a single measurement, given that the lidar systems operate basically continuously? Please discuss how you select data and discuss what criteria are used in data selection and how much averaging is done.

P13, L25 & L27. Also, the numbers in this paragraph are confusing. If there are only 42 measurements available with three wavelengths, how are there 47 samples? Of if you don't need all 3 wavelengths, then why only 47 instead of 157?

P14-15, discussion of recall, precision and accuracy (also error rate from the earlier discussion). The description of these variables could be made clearer and it should be specified that recall and precision refer to specific classes. I think the use of the terms false negative and false positive contributes to confusion rather than clarity; the descriptions are too similar to distinguish them. I think you are saying that recall for a particular class indicates the number of correct identifications divided by the number of actual instances of that class. Precision for a particular class indicates the number of correct identifications of that class divided by the number of times when that class was predicted, whether rightly or wrongly. Am I understanding it right? And what is "accuracy"? Is it defined the same as "error rate" or does "accuracy" only count the instances that are classified and not the ones that are unclassified due to overlap between classes? Are all four of these statistics independently useful, or could it be simplified by using fewer?

P15 L9 "cannot be evaluated". It seems that even though they don't appear in the training set, they were occasionally predicted by the method. If so, perhaps reporting

how often that happened would be useful.

Table 1. I don't understand what the value in parentheses represents.

Table 2. Am I interpreting this correctly, that smoke is never predicted by the classification methodology? Any comments about implications of that?

Table 2. I think PC+C in the 7th column on the top should be PC + S, and I think the 2nd CC in the 4th column on the bottom should be S.

Figure 4. What does "trained classifier" in the middle of the flowchart mean? How is it different from "typing procedure"?

———————————————————

---

## Referee Comment (RC2) · Anonymous Referee #1 · 19 Jul 2018

The manuscript 'An automatic observation-based typing method for EARLINET' by Papagiannopoulos et al. presents a classification method to determine aerosol type from intensive optical properties derived by lidar measurement. The classification scheme shall be applied to standard EARLINET measurements to expand the EARLINET data base with the corresponding aerosol type or mixture. This additional information will further enhance the value of the EARLINET data base and is thus a valuable contribution. The paper is well structured and the method and its verification clearly presented. The paper is well in the focus of ACP; I have only some minor points that have to be clarified before the publication.

Section 2.2.1 to 2.2.5: You show an extensive overview of the intensive optical properties for the different aerosol types. However, it would be valuable if you could provide

[Figure]

further information if the different intensive properties have been derived from the same studies. For using those in the classification the influence of miss-classifications and mixtures should be minimized and the measurements providing a multitude of intensive optical properties should have a larger weight. Additionally you overview mainly focuses on the information presented by Burton et al. or from EARLINET measurements. It would be also valuable to include further measurements (e.g. closer to source regions or after long-range transport) to better differentiate between possible influences of transport or mixture.

Figure 2: Why do you show profiles below the full overlap of the lidar when you do not use them for your analysis? How trustworthy are the values in these height levels? What is meant by the statement 'the layers present the same behavior'? Looking at the profiles at different wavelengths I would suggest having different behaviors at different height levels, e.g. the wavelength dependence of the backscatter coefficient, the lidar ratio and the shape of the lidar ratio between 1.8 and 2 km is different to the height range between 2 and 3.6 km, above 3.6 km the Angstroem exponent of the extinction coefficient shows a different values than below.

Figure 3: Looking at the FLEXPART footprint, can you exclude a contribution from marine aerosols?

Figure 5: An additional Figure also including the information of the depolarization ratio for the different classes would be valuable.

Figure 7: The shape of the backscatter coefficient and the extinction coefficient at 355 and at 532 nm show different shapes, but the derived profile of the lidar ratio for both wavelengths shows the same shape. What is the vertical resolution of the different profiles? Did the extinction and backscatter coefficient have the same resolution for deriving the lidar ratio?

---

## Author Comment (AC2) · 5 Sep 2018

**Referee Comment #2**

We would like to thank the anonymous referee for taking the time to carefully read the submitted paper and for commenting it. The comments were very useful for improving the readability and effectiveness of our paper. In the following, answers to comments are reported in italics, just below each related comment. The comments are underlined and numbered to ease the process. When needed, the part of the manuscript we modified or added to the old version is reported in bold. Moreover, the references are given in the end of the document.

1. Section 2.2.1 to 2.2.5: You show an extensive overview of the intensive optical properties for the different aerosol types. However, it would be valuable if you could provide further information if the different intensive properties have been derived from the same studies. For using those in the classification the influence of miss-classifications and mixtures should be minimized and the measurements providing a multitude of intensive optical properties should have a larger weight. Additionally you overview mainly focuses on the information presented by Burton et al. or from EARLINET measurements. It would be also valuable to include further measurements (e.g. closer to source regions or after long-range transport) to better differentiate between possible influences of transport or mixture.

*The proposed methodology aims to be an EARLINET stand-alone typing methodology. EARLINET collected observations, since 2000, provide an insight of the aerosol types occurring over Europe. Therefore, the methodology is set up and the aerosol classes defined. The aerosol properties are shown to vary with location and aerosol type. For example, as reported also in the paper, desert dust can have different properties depending on the source region (Arabian versus Saharan dust particles). This means that the automatic typing can be more efficient if set up at regional/continental level.*

*On this basis, Section 2.2 gives an overview of the characteristics of each aerosol type along with intensive optical properties from various literature references. The goal of this section is to introduce the aerosol classes that the automatic typing is based upon and to provide typical values of the intensive parameters. These intensive parameters correspond to the selected classifying parameters and act as a reference to our training dataset. Besides, the majority of the literature references come from EARLINET observations and reflect the variability of the aerosol properties over Europe. The overall performance of the automatic typing is based on the quality of the reference dataset and the definition of coherent aerosol classes. For the reference dataset, well-characterized EARLINET data from Pappalardo et al. (2013), Papagiannopoulos et al. (2016), and Schwarz (2016) were used. The aerosol classification follows the procedure described in Section 2.1. The aerosol classes coincide with the typical values of the Section 2.2 (see Page 27 – Table 2).*

*The algorithm has been shown to be versatile and can be adjusted to the needs of each study. The reference dataset can be enlarged with well-characterized observations and increase the number of instances of under-represented aerosol classes. In addition, new aerosol classes can be added to describe other aerosol mixtures or the aerosol classes can be redefined. For example, an aerosol class Arabian dust can be inserted in the reference dataset in order to take into account of the different characteristics of the generating desert source.*

*The next paragraph is inserted in Section 2.2:*

**As an additional consideration, the defined aerosol types presented in Sect. 2.2 may not be representative of the entire aerosol load and, furthermore, apart from the dust mixtures they do not consider other aerosol mixtures. For example, this aspect can be observed in the definition of the volcanic category where the particles have different characteristics depending the transport pattern. The particles near the source have**

**optical properties similar to desert dust whereas long-range transported volcanic plumes have the altered properties due to the sedimentation of the coarser particles. Therefore, it is important to further include a more exhaustive aerosol class analysis.**

2.  Figure 2: Why do you show profiles below the full overlap of the lidar when you do not use them for your analysis? How trustworthy are the values in these height levels? What is meant by the statement 'the layers present the same behavior'? Looking at the profiles at different wavelengths I would suggest having different behaviors at different height levels, e.g. the wavelength dependence of the backscatter coefficient, the lidar ratio and the shape of the lidar ratio between 1.8 and 2 km is different to the height range between 2 and 3.6 km, above 3.6 km the Angstroem exponent of the extinction coefficient shows a different values than below.

*The overlap height is at around 1,15 km a.s.l. for 1064 nm (Madonna et al., 2015). Values below the overlap region are not shown. We do not take into account values below the full overlap and the constructed database does not suffer from overlap issues whatsoever.*

*In the revised version of the paper we rephrased the sentence because it is actually misleading. The referee is right that three different layers are observed as reported in P5L19-20. The layer mean intensive parameters are given in Table A. The Ångström exponent for the 3 layers maintains a rather stable character with values around 0 suggesting large particles over Potenza. Regarding the lidar ratio, the values decrease with height and confirm the comment that we should not consider a single aerosol type throughout the range.*

**Table A:** *the mean intensive parameters for the 3 layers observed.*

| Layer[km] | $\kappa_\beta(355,532)$ | $\kappa_\beta(532,1064)$ | $\kappa_\beta(355,1064)$ | $\kappa_\alpha(355,532)$ | $S_{355}$ [sr] | $S_{532}$ [sr] |
|---|---|---|---|---|---|---|
| 1.6-2.0 | 0.45±0.03 | 0.32±0.03 | 0.37±0.02 | -0.2±0.2 | 53±8 | 57±8 |
| 2.0-3.5 | -0.02±0.12 | 0.42±0.06 | 0.26±0.04 | -0.3±0.2 | 48±4 | 53±4 |
| 3.5-5.0 | 0.12±0.26 | 0.42±0.18 | 0.31±0.13 | 0.4±0.2 | 46±8 | 41±5 |

*Besides, we investigated the backward trajectories for the layers of Table A using the HYSPLIT model (Stein et al., 2015). We initiated the model for a 7-day backward analysis and starting height levels the midpoints of the layers and the results are shown in Figure A. The layers 2.0-3.5 km and 3.5-5 km show a similar pattern with the air-masses flying over 2 km and originate from the Saharan desert. The layer 1.6-2.0 km follows a different pathway, the air-masses circulate over the Mediterranean Sea and Algeria and the uptake of marine particles is very likely. This information combined with the information of Table A suggests dust particles for the 2 elevated layers.*

*However, it is right to mention here that this case is reported for showing how the identification is done using intensive properties and backward trajectory analysis. All the intensive properties used for setting up the training and testing datasets are related to elevated layers separated from the planetary boundary layer and the local influence.*

[Figure]

*Figure A:* HYSPLIT backward trajectories for the aerosol layers observed over Potenza, on 14 July 2011, 19:20–22:10 UTC.

*In synthesis, the 3 layers with the synergistic use of intensive properties and backward trajectory analysis allows us to distinguish the different characteristics of the mixed dust layer in the range 1.6-2.0 km and the desert dust higher in the atmosphere (2.0-5.0 km). The intensive properties are measured with good level of uncertainty only in the layer 2.0-3.5 km. In order to avoid confusion, we will focus our comments in the range 2.0-3.5 km where the intensive property analysis is coherent and confirms the existence of a dust layer. The text implemented in P5L22-24 is given below:*

**The layer 2.0-3.5 km has a constant behavior with the range for the intensive optical profiles indicating the presence of the same type of particles. The mean values of all optical parameters in the range are calculated: lidar ratios of 48±4 sr at 355 nm and 53±4 sr at 532 nm and Ångström exponents of -0.3–0.4 were found.**

3.    Figure 3: Looking at the FLEXPART footprint, can you exclude a contribution from marine aerosols?

*The manual typing as described in Section 2.1 is not a simple issue and, of course, leaves room to questionable type assignment. The backward trajectory analysis is used synergistically with the lidar optical properties and the model outputs suffer from the high error on the path (increasing with the path itself) and the source term assignment. Therefore, the model simulations have to be used together with the observed lidar optical properties for providing a reasonable aerosol typing.*

*The FLEXPART seems to indicate the possibility of marine influence in the identified layer, however the lidar ratio values being over 50 sr indicates low or no presence of maritime aerosol particles. Furthermore, for this specific case, particle linear depolarization ratio measurements are available, however these measurements are not included in the database and therefore not reported in the submitted manuscript. Figure B shows the particle*

*linear depolarization ratio where the values are over 0.2 and, thus, confirming our hypothesis of aspherical particles.*

[Figure]

***Figure B:** Particle (blue line) and Volume (red line) linear depolarization ratio at 532 nm for the MUSA system in Potenza on 14/07/2011, 19:20–22:10 UTC.*

*In order to avoid confusion, we rephrased the sentence following the referee's comment in P5L30-31 and moved P5L24-25 to the end of it:*

**The dust-prone area of northern Africa (Morocco and northern Algeria) along with the Mediterranean Sea are most likely the sources of the observed layer and suggest a mixture of dust and marine particles. The combined information of the backward trajectory analysis and the intensive properties values indicate the presence of dust particles and they are in accordance with the typical dust values observed over Potenza (Mona et al., 2014).**

4. Figure 5: An additional Figure also including the information of the depolarization ratio for the different classes would be valuable.

*Figure C shows the characteristics of the reference dataset in terms of the depolarization ratio when plotted against the lidar ratio at 532 nm and the backscatter related Ångström exponent (355,1064) for 8 aerosol classes. The particle depolarization ratio values were assigned to the aerosol classes assuming that the intensive properties are normally distributed and using literature values as reported in P29-Table 4. Therefore, the values have no standard deviation. The figures highlight the discriminatory power of the classifying parameter among the dust-like aerosol classes, however, the particle depolarization ratio seems to have no power to separate the non-dust classes.*

[Figure]

***Figure C***: *Colored pre-specified classes and 90 % confidence ellipses for 8 aerosol classes using the classifying parameters: $\delta_{532}$, $S_{532}$, $\kappa_\beta(355,1064)$. The error bars correspond to the standard deviation of the selected mean intensive properties. CC stands for Clean Continental, D stands for dust, MD stands for mixed dust, MM stands for mixed marine, PD stands for polluted dust, PC stands for polluted continental, S stands for smoke, and V stands for volcanic particles.*

*In conjunction with the specific comment #9 of RC1, we inserted in the revised manuscript Figure D and the text below in Section 3.2.4:*

[Figure]

***Figure D:*** *Bar plots show the median (horizontal line), 25-75 percentile (box) and 5-95 percentile (whisker) of the four classifying parameters: $\delta_{532}$, $\kappa_\beta(355;1064)$, S532, and $S_{532}/S_{355}$. CC stands for Clean Continental, D stands for dust, MD stands for mixed dust, MM stands for mixed marine, PD stands for polluted dust, PC stands for polluted continental, S stands for smoke, and V stands for volcanic particles.*

**Figure 7 presents cumulative barplots with the median (black dots), the 25-75 percentile (box), the 5-95 percentile (whiskers) for all 4 classifying parameters. The figure highlights the discriminatory power of $\delta_{532}$, $\kappa_\beta(355,1064)$, and $S_{532}$, whereas the $S_{532}/S_{355}$ performs the worst. Furthermore, the figure depicts the discriminatory power of the classifying parameter among the dust-like aerosol classes, however, the particle depolarization ratio seems to have no power to separate the non-dust classes, as discussed above.**

5. Figure 7: The shape of the backscatter coefficient and the extinction coefficient at 355 and at 532 nm show different shapes, but the derived profile of the lidar ratio for both wavelengths shows the same shape. What is the vertical resolution of the different profiles? Did the extinction and backscatter coefficient have the same resolution for deriving the lidar ratio?

*P34-Figure 2 and P39-Figure 7 in the submitted paper are reporting the profiles at their highest resolutions, i.e. the particle backscatter coefficient profiles have a higher resolution when compared to the particle extinction coefficient ones and typically ultraviolet profiles have a better resolution than visible profiles. The lidar ratio profiles have the same resolution of the particle extinction coefficient profiles. Prior to the calculation of the lidar ratio, the particle backscatter coefficient profiles are smoothed using a 2nd order Savitzky-Golay filter at an effective vertical resolution that varies with height, for more details see Iarlori et al. (2015).*

*In particular, for P34-Figure 7, the vertical raw resolution of the particle backscatter coefficient profiles is 7.5 m while for the extinction coefficient the resolution varies with height. The effective resolution for 355 nm (blue line) and 532 nm (red line) is given in Figure E and follows the procedure described in Pappalardo et al. (2004). For the height range 1.2-1.9 km, the highest resolution is 240 m for 355 nm while for 532 nm varies from 240 m to 480 m. Higher in the atmosphere, the resolution degrades with height (faster for 532 nm) and becomes constant at 4.7 km for 355 nm (3.8 km for 532 nm).*

[Figure]

***Figure E:*** *The effective resolution of the extinction coefficient at 355 nm (blue line) and 532 nm (red line) for the Athens lidar on 22/05/2014, 20:28–21:28 UTC.*

*Figure F shows the same panels as for P34-Figure 7 along with the smoothed backscatter profiles (panel b) that were used for the calculation of the lidar ratio. The next sentence is implemented in the revised manuscript:*

**The particle extinction and backscatter coefficient are given with their full resolution. To calculate the lidar ratio, the backscatter coefficient was smoothed in the same effective vertical resolution using a Savitzky-Golay**

**second order filter (Iarlori et al., 2015) and only the useful range of signals was kept; the effective resolution of the resulting profiles varied from 240 m to 780 m using the method described in Pappalardo et al. (2004).**

[Figure]

*Figure F:* Optical profiles measured in Athens, on 22 May 2014, 20:28–21:28 UTC with a multiwavelength Raman lidar. From left to right, (a) the backscatter coefficient with the full resolution, (b) the smoothed backscatter coefficient, (c) the extinction coefficient, and (d) the lidar ratio. The error bars correspond to the standard deviation.

**References**

Iarlori, M., Madonna, F., Rizi, V., Trickl, T., and Amodeo, A.: Effective resolution concepts for lidar observations, Atmos. Meas. Tech., 8, 5157-5176, https://doi.org/10.5194/amt-8-5157-2015, 2015.

Madonna, F., Amato, F., Vande Hey, J., and Pappalardo, G.: Ceilometer aerosol profiling versus Raman lidar in the frame of the INTERACT campaign of ACTRIS, Atmos. Meas. Tech., 8, 2207-2223, https://doi.org/10.5194/amt-8-2207-2015, 2015.

Mona, L., Papagiannopoulos, N., Basart, S., Baldasano, J., Binietoglou, I., Cornacchia, C., and Pappalardo, G.: EARLINET dust observations vs. BSC-DREAM8b modeled profiles: 12-year-long systematic comparison at Potenza, Italy, Atmospheric Chemistry and Physics, 14, 8781–8793, doi:10.5194/acp-14-8781-2014, 2014.

Ortiz-Amezcua, P., Guerrero-Rascado, J. L., Granados-Muñoz, M. J., Benavent-Oltra, J. A., Böckmann, C., Samaras, S., Stachlewska, I. S., Janicka, Ł., Baars, H., Bohlmann, S., and Alados-Arboledas, L.: Microphysical characterization of long-range transported biomass burning particles from North America at three EARLINET stations, Atmospheric Chemistry and Physics, 17, 5931–5946, doi:10.5194/acp-17-5931-2017, 2017.

Papagiannopoulos, N., Mona, L., Alados-Arboledas, L., Amiridis, V., Baars, H., Binietoglou, I., Bortoli, D., D'Amico, G., Giunta, A., Guerrero-Rascado, J. L., Schwarz, A., Perreira, S., Spinelli, N., Wandinger, U., Wang, X., and Pappalardo, G.: CALIPSO climatological products: evaluation and suggestions from EARLINET, Atmos. Chem. Phys., 16, 2341–2357, doi:10.5194/acp-16-2341-2016, 2016a.

Pappalardo, G., Amodeo, A., Pandolfi, M., Wandinger, U., Ansmann, A., Bösenberg, J., Matthias, V., Amiridis, V., De Tomasi, F., Frioud, M., Iarlori, M., Komguem, L., Papayannis, A., Rocadenbosch, F., and Wang, X.: Aerosol lidar intercomparison in the framework of the EARLINET project. 3. Raman lidar algorithm for aerosol extinction, backscatter, and lidar ratio, Appl. Opt., 43, 5370-5385. 2004.

Pappalardo, G., Mona, L., D'Amico, G., Wandinger, U., Adam, M., Amodeo, A., Ansmann, A., Apituley, A., Alados Arboledas, L., Balis, D., Boselli, A., Bravo-Aranda, J. A., Chaikovsky, A., Comeron, A., Cuesta, J., De Tomasi, F., Freudenthaler, V., Gausa, M., Giannakaki, E., Giehl, H., Giunta, A., Grigorov, I., Groß, S., Haeffelin, M., Hiebsch, A., Iarlori, M., Lange, D., Linné, H., Madonna, F., Mattis, I., Mamouri, R.-E., McAuliffe, M. A. P., Mitev, V., Molero, F., Navas-Guzman, F., Nicolae, D., Papayannis, A., Perrone, M. R., Pietras, C., Pietruczuk, A., Pisani, G., Preißler, J., Pujadas, M., Rizi, V., Ruth, A. A., Schmidt, J., Schnell, F., Seifert, P., Serikov, I., Sicard, M., Simeonov, V., Spinelli, N., Stebel, K., Tesche, M., Trickl, T., Wang, X., Wagner, F., Wiegner, M., and Wilson, K. M.: Four-dimensional distribution of the 2010 Eyjafjallajökull volcanic cloud over Europe observed by EARLINET, Atmos. Chem. Phys., 13, 4429-4450, https://doi.org/10.5194/acp-13-4429-2013, 2013.

Schwarz, A.: Aerosol typing over Europe and its benefits for the CALIPSO and EarthCARE missions - statistical analysis based on multiwavelength aerosol lidar measurements from ground-based EARLINET stations and comparison to spaceborne CALIPSO data, Ph.D. thesis, University of Leipzig, 2016.

Stein, A.F., Draxler, R.R, Rolph, G.D., Stunder, B.J.B., Cohen, M.D., and Ngan, F.: NOAA's HYSPLIT atmospheric transport and dispersion modeling system, Bull. Amer. Meteor. Soc., 96, 2059-2077, doi:10.1175/BAMS-D-14-00110.1, 2015.

---

## Author Response (AR1)

We appreciate the Referees comments that helped us to improve the submitting paper. In the following, answers to comments are reported in italics, just below each related comment. The comments are underlined and numbered to ease the process. When needed, the part of the manuscript we modified or added to the old version is reported in bold. Moreover, the references of the cited literature are given in the end of the document.

**Referee Comment #1 (S. P. Burton)**

**Specific comments:**

1. Figures 2 and 7. What are the resolution of the extinction and backscatter profiles? How is the lidar ratio calculated? Were the extinction and backscatter at the same resolution before taking the ratio? I ask because there are differences in the shape of extinction and backscatter in each of the discussed layers that do not seem particularly consistent with the idea that each layer is a specific coherent aerosol type. For example, the "layer" below 2 km in Figure 2 has a completely different shape in backscatter vs. extinction, leading to large variability in the lidar ratio, much larger than the suggested error bars. Do you think this variability is real, or could it be that the local maximum (seen in backscatter) is smoothed out in extinction by a coarser vertical resolution? If it's real, is it likely this is a single consistent aerosol type in this layer? Similarly, the different slopes in the "layer" between 3.5 and 5 km lead to a very large slope in the lidar ratio that does not seem consistent with the idea that this is a single aerosol type. If this variability is spurious, it is liable to create additional apparent noise in the classification that is not really related to aerosol variability within classes. (Of course, spurious error would also be a concern in general, not just for classification.)

*P34-Figure 2 and P39-Figure 7 in the submitted paper are reporting the profiles at their highest resolutions, i.e. the particle backscatter coefficient profiles have a higher resolution when compared to the extinction ones and typically the ultraviolet profiles have a better resolution than the visible profiles. The lidar ratio profiles have the same resolution of the particle extinction coefficient profiles. Prior to the calculation of the lidar ratio, the particle backscatter coefficient profiles are smoothed using a 2nd order Savitzky-Golay filter at an effective vertical resolution that varies with height, for more details see Iarlori et al. (2015).*

*In particular, for P34-Figure 2, the resolution of the particle backscatter coefficient profiles is 60 m while for the particle extinction coefficient the resolution varies with height. The effective resolution for 355 nm (blue line) and 532 nm (red line) is given in Figure A and follows the procedure described in Pappalardo et al. (2004). For the height range 1.5-2.5 km, the highest resolution is 120 m for 355 nm while for 532 nm varies from 120 m to 480 m. Higher in the atmosphere, the resolution degrades with height (faster for 532 nm) and becomes constant at 4.5 km for 355 nm (3.5 km for 532 nm).*

[Figure]

*Figure A:* *The effective resolution of the extinction coefficient at 355 nm (blue line) and 532 nm (red line) for the MUSA system in Potenza on 14/07/2011, 19:20–22:10 UTC.*

*Figure B shows the same panels as for P34-Figure 2 along with the smoothed backscatter profiles (panel b) that were used for the calculation of the lidar ratio. The next lines are implemented in the revised manuscript:*

[Figure]

*Figure B:* *Optical profiles measured in Potenza, on 14 July 2011, 19:20–22:10 UTC with a multiwavelength Raman lidar. From left to right, (a) the particle backscatter coefficient with the full resolution, (b) the smoothed particle backscatter coefficient, (c) the particle extinction coefficient, and (d) the particle lidar ratio. The error bars correspond to the standard deviation.*

**The particle extinction and backscatter coefficient are given with their full resolution. To calculate the lidar ratio, the backscatter coefficient was smoothed in the same effective vertical resolution using a Savitzky-Golay second order filter (Iarlori et al., 2015) and only the useful range of signals was kept; the effective resolution of the resulting profiles varied from 120 m to 480 m using the method described in Pappalardo et al. (2004).**

*About the height-independent shape of the intensive properties, in the revised version of the paper we rephrased the sentence because it is actually misleading. The referee is right that three different layers are observed as reported in P5L19-20. The layer mean intensive parameters are given in Table A. The Ångström exponent for the 3 layers maintains a rather stable character with values around 0 suggesting large particles over Potenza. Regarding the lidar ratio, the values decrease with height and confirm the remark that we should not consider a single aerosol type throughout the range.*

**Table A:** *the mean intensive parameters for the 3 layers observed.*

| Layer [km] | $κ_β(355,532)$ | $κ_β(532,1064)$ | $κ_β(355,1064)$ | $κ_α(355,532)$ | $S_{355}$ [sr] | $S_{532}$ [sr] |
|---|---|---|---|---|---|---|
| 1.6-2.0 | 0.45±0.03 | 0.32±0.03 | 0.37±0.02 | -0.2±0.2 | 53±8 | 57±8 |
| 2.0-3.5 | -0.02±0.12 | 0.42±0.06 | 0.26±0.04 | -0.3±0.2 | 48±4 | 53±4 |
| 3.5-5.0 | 0.12±0.26 | 0.42±0.18 | 0.31±0.13 | 0.4±0.2 | 46±8 | 41±5 |

*With respect to the apparent different behavior of extinction and backscatter, we do not think that the increase of the lidar ratio below 2 km is a smoothing effect as both backscatter and extinction coefficient have the same resolution. This increase of the lidar ratio is also followed by a slight increase in the Ångström coefficient (altitudes below 1.6 km) and it might indicate the mixing with local aerosols. It is right mentioning here that below 2 km (see P33-Figure 1) the planetary boundary layer contribution has to be considered. For the tenuous layer above 3.5 km, the lidar ratio has indeed a large slope, however a few hundred meters below 5 km, and with a large statistical error because of the low aerosol concentration.*

*Besides, we investigated the backward trajectories for the layers of Table A using the HYSPLIT model (Stein et al., 2015). We initiated the model for a 7-day backward analysis and starting height levels the midpoints of the layers and the results are shown in Figure C. The layers 2.0-3.5 km and 3.5-5 km show a similar pattern with the air-masses flying over 2 km and originating source the Saharan desert. The layer 1.6-2.0 km follows a different pathway, the air-masses circulate over the Mediterranean Sea and Algeria and the uptake of marine particles is very likely. This information combined with the information of Table A suggests dust particles for the 2 elevated layers.*

*However, it is right to mention that this case (and the case in P39-Figure 7) is reported for showing how the classification is done using intensive properties and backward trajectory analyses. All the intensive properties used for setting up the reference dataset and testing dataset are related to elevated layers separated from the planetary boundary layer and local influence.*

[Figure]

*Figure C: HYSPLIT backward trajectories for the aerosol layers observed over Potenza, on 14 July 2011, 19:20–22:10 UTC.*

*In synthesis, the 3 layers with the synergistic use of intensive properties and backward trajectory analysis allows us to distinguish the different characteristics of the mixed dust layer in the range 1.6-2.0 km and the desert dust higher in the atmosphere (2.0-5.0 km). The intensive properties are measured with good level of uncertainty only in the layer 2.0-3.5 km. In order to avoid confusion, we will focus our comments in the range 2.0-3.5 km where the intensive property analysis is coherent and confirms the existence of a dust layer. The text implemented in P5L22-24 is given below:*

**The layer 2.0-3.5 km has a constant behavior with the range for the intensive optical profiles indicating the presence of the same type of particles. The mean values of all optical parameters in the range are calculated: lidar ratios of 48±4 sr at 355 nm and 53±4 sr at 532 nm and Ångström exponents ($\kappa_\beta(355,1064)$, $\kappa_\beta(532,1064)$, $\kappa_\beta(355,532)$, $\kappa_\alpha(355,532)$) of -0.3–0.4 were found.**

*Furthermore, the sentence P13L31 is changed to:*

**For the case in Sect. 2.1, the automatic algorithm labelled the aerosol layer as dust, $D_M$=1.2 and the normalized probability 55 %.**

*The same kind of discussion applies for P39-Figure 7, where the vertical raw resolution is 7.5 m and the effective resolution of the extinction coefficient profiles is given in Figure D. The resolution for both wavelengths degrades with height (faster for 532 nm). For the height range 1.2-1.9 km, the highest resolution is 240 m for 355 nm while for 532 nm varies from 240 m to 480 m. Higher in the atmosphere, the resolution degrades with height (faster for 532 nm) and becomes constant at 4.7 km for 355 nm (3.8 km for 532 nm).*

[Figure]

*Figure D:* The effective resolution of the extinction coefficient at 355 nm (blue line) and 532 nm (red line) for the Athens lidar on 22/05/2014, 20:28–21:28 UTC.

Similar to Figure B, Figure E shows the same panels as for P39-Figure 7 along with the smoothed backscatter profiles (panel b) that were used for the calculation of the lidar ratio. The next sentence is implemented in the revised manuscript:

[Figure]

*Figure E:* Optical profiles measured in Athens, on 22 May 2014, 20:28–21:28 UTC with a multiwavelength Raman lidar. From left to right, (a) the particle backscatter coefficient with the full resolution, (b) the smoothed particle backscatter coefficient, (c) the particle extinction coefficient, and (d) the particle lidar ratio. The error bars correspond to the standard deviation.

**The effective resolution of the extinction coefficient profiles varied from 240 m to 780 m using the method described in Pappalardo et al. (2004).**

2. Would you say that there is a possibility for error in the determination of "truth" aerosol types? If so, it would be good to see some discussion of that.

*Aerosol classification as described in Section 2.1 is based on a descriptive analysis of the retrieved optical properties and the model simulations, and it is a qualitative method of type assignment. Thus, there is inherent possibility of error in the determination of the true aerosol type. This error, if made, propagates into the automatic algorithm and the predicted aerosol class might deviate from the "truth" aerosol class.*

*As discussed in Sections 2.1 and 3.2, the aerosol type was decided manually by using a set of different tools (transport and trajectory models, as well as other observational data). Thus, optical properties for different aerosol types observable over Europe were derived. For the selected aerosol types, P27-Table 2 highlights the characteristics of each type and the values concur with several aerosol typing studies, such as Müller et al.(2007a), Burton et al. (2012, 2013, 2014), Gross et al. (2015), and Giannakaki et al. (2016).*

*Nevertheless, there is the possibility for error in the aerosol type definition. In supervised learning techniques, the aerosol classes must be well defined, differently the predictions of the predictive algorithm will suffer from the wrong definition. For example, this aspect can be observed in the definition of the Smoke and Volcanic category. The former refers to biomass burning particles with small size and very high lidar ratios whereas aged smoke plumes are expected to have different characteristics. The later refers only to fresh volcanic emissions and the optical characteristics are similar to the desert dust particles. Moreover, the dust type considers plumes from the Saharan desert and not the Arabian where the lidar ratio tends to be smaller. All these will lead the algorithm to misclassify the aerosol types not taken into account and allocate them to a wrong type. However, the flexibility of the algorithm allows us to circumvent it by inserting more and/or better defined aerosol types. The next lines are inserted in Section 3.2.1:*

**However, aerosol classification is based on an interpretative analysis of the retrieved optical properties and the model simulations and, it is a qualitative method of type assignment. Thus, there is inherent possibility of error in the determination of the true aerosol type. This error, if made, propagates into the automatic algorithm and the predicted aerosol class might deviate from the "truth" aerosol class.**

3. I also have a particular question about the interpretation of the influence of marine aerosol in the two case studies that are discussed at length. FLEXPART, Figures 3 and 8, seems like a very nice tool for information on aerosol source. Figure 3 seems to show a large fraction of the incidence below 2 km (a lot of the green and yellow) as being in the Mediterranean Sea, but in the discussion, no mention is made of marine influence and the case is described as "pure dust". On the other hand, in Figure 8, an apparently lesser proportion of the trajectories below 2 km are seen over the Black Sea and the Caspian Sea, and this layer is described as a mixture "enriched with marine particles during their overpass over the Black Sea". Can you clarify how we can know that there is marine influence in one case and not the other? It seems that it would be particularly difficult to say definitively when aerosol types are "pure" rather than mixed, using this method. Can you clarify whether there are additional factors that go into these judgements besides the FLEXPART tool and what uncertainties are associated with those judgements?

*The manual typing as described in Section 2.1 is not a simple issue and, of course, leaves room to questionable type assignment. The backward trajectory analysis is used synergistically with the lidar optical properties and the model outputs suffer from the high error on the path (increasing with the path itself) and the source term*

*assignment. Therefore, the model simulations have to be used together with the observed lidar optical properties for providing a reasonable aerosol typing.*

*The FLEXPART plot indicates the possibility of marine influence in the identified layer, however the lidar ratio values being over 50 sr indicates low or no presence of maritime aerosol particles. Furthermore, for this specific case, particle linear depolarization ratio measurements are available, however these measurements are not included in the database and therefore not reported in the submitted manuscript. Figure F shows the particle linear depolarization ratio where the values are over 0.2 and, thus, confirming our hypothesis of aspherical particles.*

[Figure]

**Figure F:** *Particle (blue line) and Volume (red line) linear depolarization ratio at 532 nm for the MUSA system in Potenza on 14/07/2011, 19:20–22:10 UTC.*

*In order to avoid confusion, we rephrased the sentence in P5L30-31 and moved P5L24-25 to end of it:*

**The dust-prone area of northern Africa (Morocco and northern Algeria) along with the Mediterranean Sea are most likely the sources of the observed layer and suggest a mixture of dust and marine particles. The combined information of the backward trajectory analysis and the intensive properties values indicate the presence of dust particles and they are in accordance with the typical dust values observed over Potenza (Mona et al., 2014).**

*P40-Figure 8 shows a complicated scene with multiple aerosol sources with low probability of marine influence. Unfortunately, there was a misinterpretation of the FLXEPART plot. The sentence in P14L12-15 has been rephrased and implemented in the text:*

**Therefore, the path of the air masses arriving over Athens suggests a mixture of dust and biomass burning particles, originating from the arid areas of the Aral Sea, as well as the agricultural fires in former Soviet Union countries (Papayannis et al., 2016).**

4. Section 3.2.4. When you add particle depolarization as a classification variable, I think you should still keep the original three variables. You have already shown that all three variables are sufficiently independent to be useful, so adding an independent fourth variable would be expected to produce the best classification method. I'm not following why you avoid using more than 3 variables.

*The reason that we removed a classifying parameter was that we wanted to compare in a fair and informative manner the two setups (i.e., the 3+2 and the 3+2+1). Regarding this remark, the text and figures have been corrected in the revised version of the manuscript.*

*The main features of the Figure G (same as P38-Figure 6) remain almost the same. The difference between with and without depolarization ratio of the error rate of LOOCV is bigger, thus, indicating a better performance when the 4 classifying parameters are used.*

[Figure]

***Figure G****: Bar plots showing a) the total Λ, b) the partial Λ, and c)error rate of LOOCV when comparing the training of the algorithm with (i.e., $S_{532}/S_{355}$, $S_{532}$, and $\kappa_\beta(355,1064)$) and without (i.e., $\delta_{aer,532}$, $S_{aer,532}$, and $\kappa_\beta$) particle linear depolarization values. For the partial Λ, the brown bars correspond to the backscatter-related Ångström exponent and orange one to the particle linear depolarization ratio because they represent the most significant classifying parameter in the classification.*

*Figure H (same as P41-Figure 9) shows a different behavior when compared to the graph reported in the first version of the manuscript. The prediction accuracy when depolarization is used remains almost stable (79%-85%) for all considered classes and do not increase almost monotonically when aerosol types are merged. On the contrary, for 4 classes the prediction accuracy is the lower among the other classes, however, still it is very high. The graphs and the corresponding discussion is changed accordingly.*

[Figure]

**Figure H:** Prediction accuracy for the different aerosol classes and with/without depolarization information.

*The next lines have been inserted in the revised version of the manuscript:*

*P13L6-11:* **In this case, the particle linear depolarization ratio was added to the classifying parameters. Values within the aerosol type range were randomly assigned to each sample and the Λ distribution was calculated. Total Λ is 0.004. The value of partial Λ for $\kappa_\beta$, $S_{532}$, $S_{532}/S_{355}$ and $\delta_{532}$ are 0.55, 0.34, 0.52 and 0.12 respectively. The values found for the partial Λ confirm the $\delta_{532}$ as the most important classifier for the considered dataset. For the rest aerosol groups the total and partial (for depolarization ratio) Λ are 0.005 and 0.14 respectively (7[a] classes), 0.005 and 0.12 (7[b] classes), 0.006 and 0.14 (6 classes), 0.040 and 0.68 (5 classes), and 0.050 and 0.69 (4 classes).**

*P15L12-15:* **With depolarization ratio information, the accuracy for 8 classes equals to 79 % and exceeds the 80 % for the rest aerosol classes. When comparing the accuracy of the model with and without depolarization ratio, it appears to be significantly higher until 6 classes where, further, the discrepancy diminishes (<10 %) and becomes smaller for 4 classes.**

*P16L19-21:* **Besides, the training of the algorithm with literature depolarization ratio values decreased the error rate of the leave-one-out cross validation from 24% (8 classes) to 4% (4 classes). Furthermore, the predictive accuracy increased and remained for all the aerosol classes around 80% (for 8 classes: 79%, for 7a: 81 %, for 7b: 83 %, for 6: 81 %, for 5: 85 %, and for 4: 80 %).**

5.  P4 L14 The idea that the spectral ratio of lidar ratio indicates smoke aging is still a hypothesis, based empirically on a small number of suggestive cases. Describing this relationship as "robust" overstates the case, I think. Not all of the references actually support the statement. For example, Samaras et al. 2015 have no measurements related to smoke age, but rather take it as given, substituting the spectral ratio of lidar ratios as a proxy for smoke age. Please don't use a reference to support a hypothesis that merely made use of the hypothesis (at least not without more explanation). Anyway, the current manuscript

doesn't relate to smoke aging. You could easily remove the statement and avoid controversy. At least don't overstate it and please remove references that do not really support the statement.

*The sentence has been rephrased and the non-relevant references are removed. The next sentence substitutes the phrase in P4L13-15:*

**This quantity has shown the ability to characterize the ageing status of smoke particles as well as the spectral dependence of aerosol (Müller et al., 2007a; Alados-Arboledas et al., 2011; Nepomuceno Pereira et al., 2014; Nicolae et al., 2013).**

6. Should the "clean" in the label "clean continental" be taken literally? The description of the clean continental category is obliquely defined here as a mixture of polluted continental and clean marine aerosol and indeed the data in Figure 5 also seem to support its interpretation as a mixture of the two. I think this is an interesting way to think about this type, perhaps much more useful than the standard way of thinking of it as a type that, unlike all the others, is defined by an extensive aerosol property (low aerosol loading). Any comment on this? Would a case that has the intensive properties of the clean continental class but a significant amount of aerosol optical depth (so therefore not particularly "clean") be considered "clean continental" in your analysis?

*The clean continental category, as defined in P6L20-23, presents the mixture of anthropogenic aerosols with natural sources. It is true that the clean continental subtype occupies the space between polluted continental and mixed marine (P34-Figure 5), however this aerosol type is observed in continental EARLINET sites far from maritime aerosol sources (e.g., Bucharest, Leipzig). The manual aerosol typing made by Schwarz (2016), part of the reference dataset, assigns an aerosol layer as clean continental when the aerosol concentration is low by means of optical depth. Based on this definition, the label "clean" can be taken literally. However, the automatic typing procedure does not take into account any extensive parameter, hence not "clean" aerosol layers might be classified as clean continental.*

7. P9-10. The information about Mahalanobis distance is basically repeated from earlier work. You could simplify by referencing Burton et al. 2012 and noting the different thresholds.

*The section 3.1 has been reshaped accordingly. The section reads below:*

**We developed an automated typing method, based on the work of Burton et al. (2012), but modified it in order to be compatible with the database of EARLINET. Two major steps are identified in the method proposed: the training (Sect. 3.2), and the testing (Sect. 4.1) phase. The first step consists of the following procedures. As described in Sect. 3.2.1, well characterized aerosol samples are manually separated into classes based on their physical characteristics; the set of classes constitutes the reference dataset. This procedure involves the determination of each observed aerosol layer location and the estimation of mean layer intensive optical properties. Based on this analysis, the classifying parameters that provide the required information for a better discrimination of the aerosol type are selected (Sect. 3.2.2). Next, in order to estimate how accurately a predictive model will perform, the reference dataset is split into training and validation datasets, and the application of the classifier is evaluated (Sect. 3.2.3). Sect. 3.2.4 describes the inference of characteristic depolarization values in the algorithm with the intention to increase the prediction of the model. For the second step, already pre-classified EARLINET data are used to assess the performance of the automatic typing procedure. Figure 1 illustrates the sequence of the proposed methodology starting from the setting of the training dataset, up to the assessment of the learning success during the testing phase.**

**The Mahalanobis distance of an observation from an aerosol class is estimated, and is assigned to the aerosol class for which the distance is minimum. Two screening criteria are applied to the minimum distance following the procedure of Burton et al. (2012). The methodology uses 3 and 4 classifying parameters and the minimum accepted distance for a measurement to be labelled is 4 and 4.3 respectively. Moreover, the normalized probability of the aerosol class needs to be higher than 50 %.**

8.   P10, L7. Similar probabilities for two different classes do not indicate mixing between those two classes. This only reflects that those two classes are close to each other in your measurement space. For example, any point that is close to your "smoke" class is also close to your "polluted continental" class because the classes are close to each other. If it actually is smoke plus a little bit of marine influence, the 2nd closest class will still be "polluted continental", not marine.

*Thanks for this comment. This sentence is removed.*

9.   Section 3.2.2 classifying variables selection. I think choosing variables solely based on Wilks' partial lambda may not catch everything. It may be that different variables have more power to separate different subsets of classes. For example, depolarization obviously has a lot of power to separate dust classes and almost no power to separate non-dust classes. If you had a variable that helped separate smoke from polluted continental, even partially, but did nothing else, it may have a poor partial lambda but it would nevertheless be extremely valuable. I suggest a plot similar to figure 10 in Burton et al. 2012 as a way to understand more thoroughly what each variable contributes to separating the classes. By looking at the variability of each variable within each class you can see where the overlaps are in every dimension. Your figure 5 also does this but it's usefulness maxes out at 2 dimensions, since it is very difficult to visualize more than 2 dimensions in a literal space. Your Wilks' lambda analysis suggests that the ratio of lidar ratios has significant discriminatory power, but Figure 5 does not reveal how (that is, whether some sets of classes that look like they overlap might be distinguished by the 355 nm lidar ratio or the spectral lidar ratio). It would be nice to have a visualization that answers that question. This would also address the question of whether spectral lidar ratio separates smoke and pollution aerosol (Muller et al. 2007a) which would be an interesting discussion in itself.

*The starting point of this work was a 3+2 Raman lidar configuration. Therefore, we sought for intensive parameters that are type sensitive and can be used for an effective aerosol characterization. The intensive parameters were presented in Section 2 and type specific values were given in Section 2.2. The Wilks' lambda analysis performed in Section 3.2.2 was made on the basis of the available wavelengths and the already pre-selected classifying parameters: the lidar ratio, the Ångström exponent, and the spectral ratio of the lidar ratio.*

*The bar chart in Figure I show the median (black dot), 25-75 percentile (box), and 5-95 percentile (whiskers) for each of the aerosol types and each classifying parameters. The discriminatory power of the Ångström exponent and lidar ratio is apparent whereas the ratio of the lidar ratios performs the worst among the three, in agreement with the Wilks' lambda analysis. The polluted and mixed dust yield the highest variability in the ratio of the lidar ratios that spans a wide range with median values slightly over 1. The rest of the categories have median values below 1 with dust and volcanic being the types with values almost 1. This behaviour can be seen as a confirmation of their spectral independence. Regarding Smoke and Polluted Continental there seems to be no evidence that this variable can distinguish them. However, this finding is a further confirmation of the comment #2.*

[Figure]

***Figure I***: *Bar plots and whiskers show the median (horizontal line), 25-75 percentile (box) and 5-95 percentile (whisker) of the three classifying parameters: $\kappa_\beta(355,1064)$, $S_{532}$, and $S_{532}/S_{355}$.*

10.  P16, Perhaps you could discuss more explicitly the tradeoff between more classes and less classes. All of your statistics (except Wilks' lambda) seem to show better performance with fewer classes, but that could be taken to an extreme. That is, with only one class, there would be no errors at all! How do you address this tradeoff?

*Typing in multiple classes and typing accuracy are two conflicting aspects. The number of classes as well as the typing accuracy depends on the specific needs. Currently typing methodologies provide different typing outputs with different level of accuracy: e.g., MISR (Multi-angle Imaging SpectroRadiometer) algorithm provides aerosol classification into 5 classes with high accuracy but also a finer typing (74 classes) with degraded accuracy (Kahn et al., 2015).*

*This could be an approach leaving to the specific user the possibility to select appropriate balance for his own application. Another possibility is to find a compromise between degrading accuracy and gaining insight into the aerosol type. In this direction, we indicated into the conclusions that for a 3β+2α setup, 6 aerosol classes can be used for the aerosol typing whilst for a 3β+2α+1δ setup a finer classification – with 7 aerosol classes – can be made. Whenever better performance in terms of accuracy is requested the number of aerosol classes can be reduced to 4 because of the similarities of the physical characteristics of the aerosol classes. The choice of 4 classes is also a good analogy for the proposed aerosol typing by Baars et al. (2017).*

*The next phrase is inserted in Section 3.2.3:*

**It should be mentioned that the typing in multiple classes and typing accuracy are two conflicting aspects. The choice of 8 aerosol classes appears to be sufficient to describe the major aerosol components, however ostentatious for a 3+2 lidar configuration. 4 classes, on the other hand, provide a coarse aerosol characterization and the prediction accuracy of the algorithm is expected to increase.**

11. Do you plan to share these aerosol typing results publically? What about the training database of manually typed samples? Also please include links to the EARLINET database.

*The data used into this paper are freely and publicly available on the European Research Infrastructure for the observation of Aerosol, Clouds, and Trace gases (ACTRIS) website (https://www.actris.eu/). Moreover, the complete EARLINET dataset for 2000-2015 period is currently on the way for publication on the CERA database. This information will be added, if available, before the final publication of the paper.*

*Additionally, the reference and training datasets will be publicly available as well: a special dataset will be available through EARLINET/ACTRIS web page. In addition, this dataset will be part of a larger initiative about aerosol typing within AEROSAT (International Satellite Aerosol Science Network; http://www.aero-sat.org/), in which it is planned to set up an extensive reference dataset for aerosol typing investigation.*

**Typos and requests for clarification:**

1. P2, last sentence: Both cluster analysis techniques and supervised classification techniques need the number of groups as input.

*Yes, it is true that for cluster analysis it is needed the number of groups as input, however the identities of groups are not known in advance in contrast with the classification analysis (see Wilks 2006, page 529). The P2L32-33 sentence is rephrased:*

**Whereas in cluster analysis, the groups are not known beforehand and the classifier is tasked with it.**

2. P4 L7: "operates" should be "operate"

*Ok, fixed.*

3. P5, L18: talks about the region of incomplete overlap. What altitude does this go up to?

*The full overlap height for the 1064 nm channel is approximately 405 a.g.l. m (Madonna et al., 2015). This information is implemented in the text:*

**MUSA has a full overlap at around 1,15 km a.s.l. for 1064 nm (Madonna et al., 2015).**

4. P5, L22-23: the sentence "the aforementioned layers" is unclear and should be reworded. I think you are suggesting that the intensive properties are approximately constant throughout each layer, but that does not really appear to be true, so perhaps I'm misunderstanding the wording.

*Based on the comment #1, the sentences P5L22-24 now read:*

**The layer 2.0-3.5 km has a constant behavior with the range for the intensive optical profiles indicating the presence of the same type of particles. The mean values of all optical parameters in the range are calculated: lidar ratios of 48±4 sr at 355 nm and 53±4 sr at 532 nm and Ångström exponents ($k_\beta$(355,1064), $k_\beta$(532,1064), $k_\beta$(355,532), $k_\alpha$(355,532)) of -0.3–0.4 were found.**

5. P5, L24 and throughout: there are four Angstrom exponents discussed but often the text refers to "Angstrom exponent" as if there is only one. Please clarify which one you mean. Likewise, it should be specified which wavelength is meant when "lidar ratio" is used.

*For P5L24, we use "Ångström exponents" for all the available 4 sets appearing in Figure 2 – $\kappa_\beta$(355,1064), $\kappa_\beta$(532,1064), $\kappa_\beta$(355,532), $\kappa_\alpha$(355,532). The Ångström exponent and lidar ratio wavelengths, whenever it was needed, are now reported. The following sentences are implemented in the text.*

*P8L23:* **lidar ratios at 532 nm of 50–65 sr**
*P8L27-28*: **Pappalardo et al. (2004) and Wang et al. (2008) reported lidar ratios of 50–60 sr at 355 nm and backscatter-related Ångström exponent (355, 532) of 2.4.**
*P5L24:* **Ångström exponents ($\kappa_\beta$(355,1064), $\kappa_\beta$(532,1064), $\kappa_\beta$(355,532), $\kappa_\alpha$(355,532)) of 0–0.4 were found.**
*P10L28:* **Ångström exponent for the available.**
*P10L30:* **The Ångström exponents ($\kappa_\beta$(355,1064), $\kappa_\beta$(532,1064), $\kappa_\beta$(355,532), $\kappa_\alpha$(355,532)) for mixed dust are.**
*P10L31:* **Ångström exponents lie.**
*P10L33*: **with mean Ångström exponent from all the available variables around ~1.4 and ~1.3 respectively.**
*P11L1:* **81±16 sr and 78±11 sr for 355 nm and 532 nm respectively.**
*P11L3:* **Ångström exponents ($\kappa_\beta$(355,1064), $\kappa_\beta$(532,1064), $\kappa_\beta$(355,532), $\kappa_\alpha$(355,532)) are.**
*P11L5:* **Ångström exponents ($\kappa_\beta$(355,1064), $\kappa_\beta$(532,1064), $\kappa_\beta$(355,532), $\kappa_\alpha$(355,532)) in the range 0.8–1.0**
*P12L3:* **$S_{532}$ and $\kappa_\beta$(355,1064)**
*P13L16:* **lidar ratio at 532 nm.**
*P14L5:* **Within this layer the mean value of backscatter-related Ångström (355,1064) exponent is 0.9±0.1**
*P38-Figure 6:* **$\kappa_\beta$(355,1064).**

6. P6 L23. I recommend taking more care about using the word "absorbing". It seems that "more absorbing" is here used as a synonym for "higher lidar ratio", but lidar ratio depends on particle size and other factors as well as light absorption and is really not as direct an indicator as this language suggests. Also P11 L4 and L14 (continental pollution is not necessarily absorbing); P8 L5 (smoke is often absorbing but not always highly absorbing especially when aged); and perhaps elsewhere.

*It is true that the lidar ratio depends on many factors and it is not a direct indicator of light absorption. The following sentences are changed and implemented in the manuscript.*

*P6L23:* **The clean continental, therefore, differentiates from the polluted continental type due to lower lidar ratio values.**

*P11L4:* **This characteristic separates clean continental from polluted continental as the particles yield lower lidar ratio values.**

*P11L14:* **Two pathways were followed, first, the smoke and the polluted continental categories were grouped into the more generic type of small with high lidar ratio values.**

*P8L5:* **Generally, smoke particles are relatively small and spherical that produce low depolarization, high Ångström exponents, and large lidar ratios (Amiridis et al., 2009; Baars et al., 2012; Giannakaki et al., 2016).**

7. P7, L14. Delete "mainly". Although the variability in the lidar ratio is a "hot topic" there is also significant variability in, for instance, extinction Angstrom exponent.

*Deleted.*

8. P7, L31. The suggestion for CALIPSO to add a dust+marine type was made several times before 2016 also, for example Kim et al. 2013, Burton et al. 2013, Rogers et al. 2014.

*Thank you for this suggestion. These literature references are implemented in the document and the sentence reads as follows:*

**Several studies (Burton et al., 2013; Kim et al., 2013; Rogers et al., 2014; Papagiannopoulos et al., 2016a) have indicated that this mixture is important and suggested its inclusion in the CALIPSO retrieval scheme for improving the accuracy of aerosol backscatter and extinction coefficient profiles.**

9. P8 L10 and in the references: I think Pereira should be Nepomucino Pereira (that is, the first author appears to have a two-part surname).

*Yes, it is fixed. The reference becomes:*

**Nepomuceno Pereira, S., Preißler, J., Guerrero-Rascado, J. L., Silva, A. M., and Wagner, F.: Forest Fire Smoke Layers Observed in the Free Troposphere over Portugal with a Multiwavelength Raman Lidar: Optical and Microphysical Properties, Scientific World Journal, 2014, doi:10.1155/2014/42183, 2014.**

10. P9 first paragraph and third paragraph: There are a few places, including these 2 paragraphs, where the wording is awkward with several errors in English language usage that make them hard to understand. Please reword for clarity

*Following specific comment #7, the first paragraph is reworded and merged with the second paragraph as follows:*

**We developed an automated typing method, based on the work of Burton et al. (2012), but modified it in order to be compatible with the database of EARLINET. Two major steps are identified in the method proposed: the training (Sect. 3.2), and the testing (Sect. 4.1) phase. The first step consists of the following procedures. As described in Sect. 3.2.1, well characterized aerosol samples are manually separated into classes based on their physical characteristics; the set of classes constitutes the reference dataset. This procedure involves the determination of each observed aerosol layer location and the estimation of mean layer intensive optical properties. Based on this analysis, the classifying parameters that provide the required information for a better discrimination of the aerosol type are selected (Sect. 3.2.2). Next, in order to estimate how accurately a predictive model will perform, the reference dataset is split into training and validation datasets, and the application of the classifier is evaluated (Sect. 3.2.3). Sect. 3.2.4 describes the inference of characteristic depolarization values in the algorithm with the intention to increase the prediction of the model. For the second step, already pre-classified EARLINET data are used to assess the performance of the automatic typing procedure. Figure 1 illustrates the sequence of the proposed methodology starting from the setting of the training dataset, up to the assessment of the learning success during the testing phase.**

*The third paragraph is changed:*

**For the second step, unclassified EARLINET data (the testing dataset) is categorized using the reference dataset. Besides, the testing dataset has been classified following the method shown in Sect. 2.1 and, hence, compared against the output of the automatic procedure. Figure 4 illustrates the sequence of the proposed methodology, starting from the setting of the training dataset up to the assessment of the learning success during the testing phase.**

11. P12 L28, I think you mean Burton et al. 2015, not 2014. Burton, S. P., Hair, J. W., Kahnert, M., Ferrare, R. A., Hostetler, C. A., Cook, A. L., Harper, D. B., Berkoff, T. A., Seaman, S. T., Collins, J. E., Fenn, M. A., and Rogers, R. R.: Observations of the spectral dependence of linear particle depolarization ratio of aerosols using NASA Langley airborne High Spectral Resolution Lidar, Atmos. Chem. Phys., 15, 13453- 13473, 10.5194/acp-15-13453-2015, 2015.

*Thank you. Fixed.*

12. P13, L25 & L27, and throughout. When you count measurements or samples, what defines a single measurement, given that the lidar systems operate basically continuously? Please discuss how you select data and discuss what criteria are used in data selection and how much averaging is done.

*EARLINET is a research network born in 2000 on the basis of bringing together the existing research lidars around Europe. At that time, none of the members could provide continuous measurements. Nowadays, there are some systems working 24/7 unattended, but EARLINET is working on a measurements schedule (see P4L1). Furthermore, measurements are performed during CALIPSO site overpasses and special events, such as biomass burning episodes, dust advection, and volcanic eruptions. To accommodate these kind of events, it was agreed to have some measurements lasting 2-3 hours in order to monitor the evolution of each specific event. Then, it is responsibility of the stations to provide aerosol profiles better representing the observed atmospheric aerosol conditions: i.e., deciding the time window and the time averaging needed for the lidar signal inversion.*

*Throughout the paper, we refer to measurements as a set of aerosol optical profiles reported in the EARLINET database and corresponding to the same temporal window, which typically spans about 1 hour of temporal integration.*

*For making clearer this point we added the following sentence in P4L6:*

**Throughout the paper, we refer to measurements as a set of aerosol optical profiles reported in the EARLINET database that correspond to the same temporal window, which typically extends for about 1 hour.**

*We refer instead to samples as identified layers. We changed samples into layers in the related text.*

13. P13, L25 & L27. Also, the numbers in this paragraph are confusing. If there are only 42 measurements available with three wavelengths, how are there 47 samples? Of if you don't need all 3 wavelengths, then why only 47 instead of 157?

*For the considered period there are 157 measurements (see definition above). 42 of which provide 3 (at 355 nm, 532 nm, and 1064 nm) backscatter coefficient profiles and 2 (355 nm and 532 nm) extinction coefficient profiles (this reduced number is due to low SNR for some optical properties especially extinction with inhibited the retrieval but most to the fact that in daytime conditions Raman technique does not allow the extinction retrieval at all). Out of the 42 measurements, 47 aerosol layers have been identified in the free troposphere and a reasonable typing was possible.*

14. P14-15, discussion of recall, precision and accuracy (also error rate from the earlier discussion). The description of these variables could be made clearer and it should be specified that recall and precision refer to specific classes. I think the use of the terms false negative and false positive contributes to confusion rather than clarity; the descriptions are too similar to distinguish them. I think you are saying that recall for a particular class indicates the number of correct identifications divided by the number of actual instances of that class. Precision for a particular class indicates the number of correct identifications of that class divided by the number of times when that class was predicted, whether rightly or wrongly.

Am I understanding it right? And what is "accuracy"? Is it defined the same as "error rate" or does "accuracy" only count the instances that are classified and not the ones that are unclassified due to overlap between classes? Are all four of these statistics independently useful, or could it be simplified by using fewer?

Am I understanding it right? *Yes, this is what recall and precision are.*

And what is "accuracy"? Is it defined the same as "error rate" or does "accuracy" only count the instances that are classified and not the ones that are unclassified due to overlap between classes? Accuracy is defined as 1-Error Rate (as introduced in Section 3.2.3), and it is demonstrating how often the classifier is correct.

Are all four of these statistics independently useful, or could it be simplified by using fewer? *In order to reduce the number of the statistics, the accuracy is replaced by the error rate throughout the document.*

*The description of the error rate in P12L20-21 is changed and implemented in the text:*

**The error rate is estimated as a percentage of all incorrect predictions divided by the total number of the reference dataset, and is equivalent to 1 minus accurary. Values near zero show high predictive performance while values near one show low predictive performance.**

*The recall and precision relate to one class independent of any other classes. Error rate or accuracy is not a reliable metric for the assessment of the real performance of the classifier, because it may produce misleading results. That is, the number of observations vary greatly and this effect can be seen in P30-Table 5. We decided to keep the metrics recall and precision, however we provide a clearer description and the formulas removed. The P14L25-30 & P15L1-8 is restructured and the next sentences are introduced in the document:*

**… is already known. Recall of an aerosol group is defined as the number of correctly predicted cases over the number of correctly plus the number of incorrectly predicted cases. Recall can be thought as the model's ability to predict the specific aerosol class. Precision of an aerosol class is defined as the number of correctly predicted cases over the number of correctly predicted cases plus the number of incorrectly predicted cases that belong to this aerosol class. In other words, given the prediction of a specific class, what is the probability of being correct?**

15. P15 L9 "cannot be evaluated". It seems that even though they don't appear in the training set, they were occasionally predicted by the method. If so, perhaps reporting ow often that happened would be useful.

*This comment refers to P15L30. The frequency of detection for MD (PD) is 18% (4%) for 8 classes, 15% (3%) for $7^b$ classes, 17% (3%) for $7^a$ classes, and 15% (3%) when 3 classifying parameters are used. The algorithm predicted as MD only dust cases with mean $S_{532}=47\pm2$ sr, $S_{532}/S_{355}=1.1\pm0.1$ and $\kappa_\beta(355,1064)=0.4\pm0.1$; thus the lower lidar ratio and the over 1 value of the ratio of the lidar ratio assigns these cases as dust. The PD case refers to a PC case with $S_{532}=69$ sr, $S_{532}/S_{355}=1.2$ and $\kappa_\beta(355,1064)=0.9$.*

*The frequency of detection for MD is 0% for all classes when particle depolarization ratio is added. The frequency for PD is 17% for 8 classes, 13% for $7^b$ classes, 17% for $7^a$ classes, and 13% for 6 classes. The PD refers to a case with low particle depolarization ratio. Following this remark, we added the next sentences in the revised document:*

**Note that mixed dust and polluted dust aerosol types are not reported in the tables due to the fact that they are not present in Table 5 and these parameters cannot be evaluated. The frequency of detection for MD (PD) is 18% (4%) for 8 classes, 15% (3%) for $7^b$ classes, 17% (3%) for $7^a$ classes, and 15% (3%) when 3 classifying**

**parameters are used. The algorithm predicted as MD only dust cases with $S_{532}$ around 45 sr and $S_{532}/S_{355}$ over 1. The PD case refers to a PC case with $\kappa_\beta(355,1064)$ lower than 1. The frequency of detection for MD is 0% for all classes when depolarization ratio is added. The frequency for PD is 17% for 8 classes, 13% for $7^b$ classes, 17% for $7^a$ classes, and 13% for 6 classes. The wrongly classified cases have depolarization ratio around 20%.**

16. Table 1. I don't understand what the value in parentheses represents.

*The number in parentheses indicate the mixtures of more than two aerosol types. These numbers will be removed because they do not add to our discussion.*

17. Table 2. Am I interpreting this correctly, that smoke is never predicted by the classification methodology? Any comments about implications of that?

*We think the comment refers to P31-Table 6. The smoke cases examined show optical properties similar to polluted continental, hence the Mahalanobis distance classifier labels them as polluted continental. The mean lidar ratio at 532 nm is 61±5, the β-related Ångström coefficient (355, 1064 nm) is 1.5±0.2, and the spectral ratio of the lidar ratios is 1.15±0.08.*

*We think that the high values of lidar ratio considered in the reference dataset do not allow the correct classification of smoke layers and as a consequence these smoke layers are typed as polluted continental. We acknowledged in the manuscript that well-defined aerosol types are paramount for the correct classification and, for instance, it is needed to split the smoke category into fresh and aged biomass burning.*

18. Table 2. I think PC+C in the 7th column on the top should be PC + S, and I think the 2nd CC in the 4th column on the bottom should be S.

*This was a typo and it is fixed.*

19. Figure 4. What does "trained classifier" in the middle of the flowchart mean? How is it different from "typing procedure"?

*By trained classifier we consider the whole iterative process whereby we build the best possible classifier, i.e., the Section 3. Whereas the typing procedure refers to the labeling of the instances using the Mahalanobis distance classifier.*

*We consider changing the word "classifier" with "classifying parameter" in the P13L6 and P13L9 as it might confuse the reader.*

**Referee Comment #2**

1. Section 2.2.1 to 2.2.5: You show an extensive overview of the intensive optical properties for the different aerosol types. However, it would be valuable if you could provide further information if the different intensive properties have been derived from the same studies. For using those in the classification the influence of miss-classifications and mixtures should be minimized and the measurements providing a multitude of intensive optical properties should have a larger weight. Additionally you overview mainly focuses on the information presented by Burton et al. or from EARLINET measurements. It would be also valuable to include further measurements (e.g. closer to source regions or after long-range transport) to better differentiate between possible influences of transport or mixture.

*The proposed methodology aims to be an EARLINET stand-alone typing methodology. EARLINET collected observations, since 2000, provide an insight of the aerosol types occurring over Europe. Therefore, the methodology is set up and the aerosol classes defined. The aerosol properties are shown to vary with location and aerosol type. For example, as reported also in the paper, desert dust can have different properties depending on the source region (Arabian versus Saharan dust particles). This means that the automatic typing can be more efficient if set up at regional/continental level.*

*On this basis, Section 2.2 gives an overview of the characteristics of each aerosol type along with intensive optical properties from various literature references. The goal of this section is to introduce the aerosol classes that the automatic typing is based upon and to provide typical values of the intensive parameters. These intensive parameters correspond to the selected classifying parameters and act as a reference to our training dataset. Besides, the majority of the literature references come from EARLINET observations and reflect the variability of the aerosol properties over Europe. The overall performance of the automatic typing is based on the quality of the reference dataset and the definition of coherent aerosol classes. For the reference dataset, well-characterized EARLINET data from Pappalardo et al. (2013), Papagiannopoulos et al. (2016), and Schwarz (2016) were used. The aerosol classification follows the procedure described in Section 2.1. The aerosol classes coincide with the typical values of the Section 2.2 (see Page 27 – Table 2).*

*The algorithm has been shown to be versatile and can be adjusted to the needs of each study. The reference dataset can be enlarged with well-characterized observations and increase the number of instances of under-represented aerosol classes. In addition, new aerosol classes can be added to describe other aerosol mixtures or the aerosol classes can be redefined. For example, an aerosol class Arabian dust can be inserted in the reference dataset in order to take into account of the different characteristics of the generating desert source.*

*The next paragraph is inserted in Section 2.2:*

**As an additional consideration, the defined aerosol types presented in Sect. 2.2 may not be representative of the entire aerosol load and, furthermore, apart from the dust mixtures they do not consider other aerosol mixtures. For example, this aspect can be observed in the definition of the volcanic category where the particles have different characteristics depending the transport pattern. The particles near the source have optical properties similar to desert dust whereas long-range transported volcanic plumes have the altered properties due to the sedimentation of the coarser particles. Therefore, it is important to further include a more exhaustive aerosol class analysis.**

2. Figure 2: Why do you show profiles below the full overlap of the lidar when you do not use them for your analysis? How trustworthy are the values in these height levels? What is meant by the statement 'the layers present the same behavior'? Looking at the profiles at different wavelengths I would suggest having

different behaviors at different height levels, e.g. the wavelength dependence of the backscatter coefficient, the lidar ratio and the shape of the lidar ratio between 1.8 and 2 km is different to the height range between 2 and 3.6 km, above 3.6 km the Angstroem exponent of the extinction coefficient shows a different values than below.

The overlap height is at around 1,15 km a.s.l. for 1064 nm (Madonna et al., 2015). Values below the overlap region are not shown. We do not take into account values below the full overlap and the constructed database does not suffer from overlap issues whatsoever.

In the revised version of the paper we rephrased the sentence because it is actually misleading. The referee is right that three different layers are observed as reported in P5L19-20. The layer mean intensive parameters are given in Table A. The Ångström exponent for the 3 layers maintains a rather stable character with values around 0 suggesting large particles over Potenza. Regarding the lidar ratio, the values decrease with height and confirm the comment that we should not consider a single aerosol type throughout the range.

**Table B:** the mean intensive parameters for the 3 layers observed.

| Layer[km] | $\kappa_\beta(355,532)$ | $\kappa_\beta(532,1064)$ | $\kappa_\beta(355,1064)$ | $\kappa_\alpha(355,532)$ | $S_{355}$ [sr] | $S_{532}$ [sr] |
|-----------|-------------------------|--------------------------|--------------------------|--------------------------|----------------|----------------|
| 1.6-2.0   | 0.45±0.03               | 0.32±0.03                | 0.37±0.02                | -0.2±0.2                 | 53±8           | 57±8           |
| 2.0-3.5   | -0.02±0.12              | 0.42±0.06                | 0.26±0.04                | -0.3±0.2                 | 48±4           | 53±4           |
| 3.5-5.0   | 0.12±0.26               | 0.42±0.18                | 0.31±0.13                | 0.4±0.2                  | 46±8           | 41±5           |

Besides, we investigated the backward trajectories for the layers of Table A using the HYSPLIT model (Stein et al., 2015). We initiated the model for a 7-day backward analysis and starting height levels the midpoints of the layers and the results are shown in Figure A. The layers 2.0-3.5 km and 3.5-5 km show a similar pattern with the air-masses flying over 2 km and originate from the Saharan desert. The layer 1.6-2.0 km follows a different pathway, the air-masses circulate over the Mediterranean Sea and Algeria and the uptake of marine particles is very likely. This information combined with the information of Table A suggests dust particles for the 2 elevated layers.

However, it is right to mention here that this case is reported for showing how the identification is done using intensive properties and backward trajectory analysis. All the intensive properties used for setting up the training and testing datasets are related to elevated layers separated from the planetary boundary layer and the local influence.

[Figure]

*Figure J: HYSPLIT backward trajectories for the aerosol layers observed over Potenza, on 14 July 2011, 19:20–22:10 UTC.*

*In synthesis, the 3 layers with the synergistic use of intensive properties and backward trajectory analysis allows us to distinguish the different characteristics of the mixed dust layer in the range 1.6-2.0 km and the desert dust higher in the atmosphere (2.0-5.0 km). The intensive properties are measured with good level of uncertainty only in the layer 2.0-3.5 km. In order to avoid confusion, we will focus our comments in the range 2.0-3.5 km where the intensive property analysis is coherent and confirms the existence of a dust layer. The text implemented in P5L22-24 is given below:*

**The layer 2.0-3.5 km has a constant behavior with the range for the intensive optical profiles indicating the presence of the same type of particles. The mean values of all optical parameters in the range are calculated: lidar ratios of 48±4 sr at 355 nm and 53±4 sr at 532 nm and Ångström exponents ($\kappa_\beta$(355,1064), $\kappa_\beta$(532,1064), $\kappa_\beta$(355,532), $\kappa_\alpha$(355,532)) of -0.3–0.4 were found.**

3.  Figure 3: Looking at the FLEXPART footprint, can you exclude a contribution from marine aerosols?

*The manual typing as described in Section 2.1 is not a simple issue and, of course, leaves room to questionable type assignment. The backward trajectory analysis is used synergistically with the lidar optical properties and the model outputs suffer from the high error on the path (increasing with the path itself) and the source term assignment. Therefore, the model simulations have to be used together with the observed lidar optical properties for providing a reasonable aerosol typing.*

*The FLEXPART seems to indicate the possibility of marine influence in the identified layer, however the lidar ratio values being over 50 sr indicates low or no presence of maritime aerosol particles. Furthermore, for this specific case, particle linear depolarization ratio measurements are available, however these measurements are not*

*included in the database and therefore not reported in the submitted manuscript. Figure B shows the particle linear depolarization ratio where the values are over 0.2 and, thus, confirming our hypothesis of aspherical particles.*

[Figure]

**Figure K:** *Particle (blue line) and Volume (red line) linear depolarization ratio at 532 nm for the MUSA system in Potenza on 14/07/2011, 19:20–22:10 UTC.*

*In order to avoid confusion, we rephrased the sentence following the referee's comment in P5L30-31 and moved P5L24-25 to the end of it:*

**The dust-prone area of northern Africa (Morocco and northern Algeria) along with the Mediterranean Sea are most likely the sources of the observed layer and suggest a mixture of dust and marine particles. The combined information of the backward trajectory analysis and the intensive properties values indicate the presence of dust particles and they are in accordance with the typical dust values observed over Potenza (Mona et al., 2014).**

4. Figure 5: An additional Figure also including the information of the depolarization ratio for the different classes would be valuable.

*Figure C shows the characteristics of the reference dataset in terms of the depolarization ratio when plotted against the lidar ratio at 532 nm and the backscatter related Ångström exponent (355,1064) for 8 aerosol classes. The particle depolarization ratio values were assigned to the aerosol classes assuming that the intensive properties are normally distributed and using literature values as reported in P29-Table 4. Therefore, the values have no standard deviation. The figures highlight the discriminatory power of the classifying parameter among the dust-like aerosol classes, however, the particle depolarization ratio seems to have no power to separate the non-dust classes.*

[Figure]

*Figure L*: *Colored pre-specified classes and 90 % confidence ellipses for 8 aerosol classes using the classifying parameters: $\delta_{532}$, $S_{532}$, $\kappa_\beta(355,1064)$. The error bars correspond to the standard deviation of the selected mean intensive properties. CC stands for Clean Continental, D stands for dust, MD stands for mixed dust, MM stands for mixed marine, PD stands for polluted dust, PC stands for polluted continental, S stands for smoke, and V stands for volcanic particles.*

*In conjunction with the specific comment #9 of RC1, we inserted in the revised manuscript Figure D and the text below in Section 3.2.4:*

[Figure]

*Figure M: Bar plots show the median (horizontal line), 25-75 percentile (box) and 5-95 percentile (whisker) of the four classifying parameters: $\delta_{532}$, $\kappa_\beta(355;1064)$, $S532$, and $S_{532}/S_{355}$. CC stands for Clean Continental, D stands for dust, MD stands for mixed dust, MM stands for mixed marine, PD stands for polluted dust, PC stands for polluted continental, S stands for smoke, and V stands for volcanic particles.*

**Figure 7 presents cumulative bar plots with the median (black dots), the 25-75 percentile (box), the 5-95 percentile (whiskers) for all 4 classifying parameters. The figure highlights the discriminatory power of $\delta_{532}$, $\kappa_\beta(355,1064)$, and $S_{532}$, whereas the $S_{532}/S_{355}$ performs the worst. Furthermore, the figure depicts the discriminatory power of the classifying parameter among the dust-like aerosol classes, however, the particle depolarization ratio seems to have no power to separate the non-dust classes, as discussed above.**

5.    Figure 7: The shape of the backscatter coefficient and the extinction coefficient at 355 and at 532 nm show different shapes, but the derived profile of the lidar ratio for both wavelengths shows the same shape. What is the vertical resolution of the different profiles? Did the extinction and backscatter coefficient have the same resolution for deriving the lidar ratio?

*P34-Figure 2 and P39-Figure 7 in the submitted paper are reporting the profiles at their highest resolutions, i.e. the particle backscatter coefficient profiles have a higher resolution when compared to the particle extinction coefficient ones and typically ultraviolet profiles have a better resolution than visible profiles. The lidar ratio profiles have the same resolution of the particle extinction coefficient profiles. Prior to the calculation of the lidar ratio, the particle backscatter coefficient profiles are smoothed using a 2nd order Savitzky-Golay filter at an effective vertical resolution that varies with height, for more details see Iarlori et al. (2015).*

*In particular, for P34-Figure 7, the vertical raw resolution of the particle backscatter coefficient profiles is 7.5 m while for the extinction coefficient the resolution varies with height. The effective resolution for 355 nm (blue line) and 532 nm (red line) is given in Figure E and follows the procedure described in Pappalardo et al. (2004). For the height range 1.2-1.9 km, the highest resolution is 240 m for 355 nm while for 532 nm varies from 240 m to 480 m. Higher in the atmosphere, the resolution degrades with height (faster for 532 nm) and becomes constant at 4.7 km for 355 nm (3.8 km for 532 nm).*

[Figure]

***Figure N:** The effective resolution of the extinction coefficient at 355 nm (blue line) and 532 nm (red line) for the Athens lidar on 22/05/2014, 20:28–21:28 UTC.*

*Figure F shows the same panels as for P34-Figure 7 along with the smoothed backscatter profiles (panel b) that were used for the calculation of the lidar ratio. The next sentence is implemented in the revised manuscript:*

**The effective resolution of the extinction coefficient profiles varied from 240 m to 780 m using the method described in Pappalardo et al. (2004).**

[Figure]

***Figure O:*** *Optical profiles measured in Athens, on 22 May 2014, 20:28–21:28 UTC with a multiwavelength Raman lidar. From left to right, (a) the backscatter coefficient with the full resolution, (b) the smoothed backscatter coefficient, (c) the extinction coefficient, and (d) the lidar ratio. The error bars correspond to the standard deviation.*

[revised manuscript text omitted]